# Extracellular electron transfer increases fermentation in lactic acid bacteria via a hybrid metabolism

Sara Tejedor-Sanz[1,2†], Eric T Stevens[3†], Siliang Li[1], Peter Finnegan[3], James Nelson[4], Andre Knoesen[4], Samuel H Light[5], Caroline M Ajo-Franklin[1,2*], Maria L Marco[3*]

[1]Department of BioSciences, Rice University, Houston, United States; [2]Biological Nanostructures Facility, The Molecular Foundry, Lawrence Berkeley National Laboratory, Berkeley, United States; [3]Department of Food Science & Technology, University of California-Davis, Davis, United States; [4]Department of Electrical and Computer Engineering, University of California-Davis, Davis, United States; [5]Department of Microbiology, University of Chicago, Chicago, United States

*For correspondence:
cajo-franklin@rice.edu (CMA-F);
mmarco@ucdavis.edu (MLM)

†These authors contributed equally to this work

Competing interest: The authors declare that no competing interests exist.

**Abstract** Energy conservation in microorganisms is classically categorized into respiration and fermentation; however, recent work shows some species can use mixed or alternative bioenergetic strategies. We explored the use of extracellular electron transfer for energy conservation in diverse lactic acid bacteria (LAB), microorganisms that mainly rely on fermentative metabolism and are important in food fermentations. The LAB *Lactiplantibacillus plantarum* uses extracellular electron transfer to increase its $NAD^+$/NADH ratio, generate more ATP through substrate-level phosphorylation, and accumulate biomass more rapidly. This novel, hybrid metabolism is dependent on a type-II NADH dehydrogenase (Ndh2) and conditionally requires a flavin-binding extracellular lipoprotein (PplA) under laboratory conditions. It confers increased fermentation product yield, metabolic flux, and environmental acidification in laboratory media and during kale juice fermentation. The discovery of a single pathway that simultaneously blends features of fermentation and respiration in a primarily fermentative microorganism expands our knowledge of energy conservation and provides immediate biotechnology applications.

## Editor's evaluation

In this study, the authors describe unique metabolic strategies, including extracellular electron transfer, utilized by the lactic acid bacterium *Lactiplantibacillus plantarum*. The ability to shift and/or accelerate metabolism of lactic acid bacteria capable of extracellular electron transfer may have interesting biotechnological applications.

## Introduction

The ways in which microorganisms extract energy to maintain cellular functions are directly linked to their environment, including the availability of nutrients and cooperative or antagonistic interactions with other organisms (*Haruta and Kanno, 2015*). Microorganisms must also maintain redox homeostasis by responding to oxidative and reductive changes inside and outside the cell (*Sporer et al., 2017*). Ultimately, microorganisms that can effectively generate cellular energy while also managing redox requirements will maintain higher growth and survival rates, and therefore exhibit greater ecological fitness.

**eLife digest** Bacteria produce the energy they need to live through two processes, respiration and fermentation. While respiration is often more energetically efficient, many bacteria rely on fermentation as their sole means of energy production. Respiration normally depends on the presence of small soluble molecules, such as oxygen, that can diffuse inside the cell, but some bacteria can use metals or other insoluble compounds found outside the cell to perform 'extracellular electron transfer'.

Lactic acid bacteria are a large group of bacteria that have several industrial uses and live in many natural environments. These bacteria survive using fermentation, but they also carry a group of genes needed for extracellular electron transfer. It is unclear whether they use these genes for respiration or if they have a different purpose.

Tejedor-Sanz, Stevens et al. used a lactic acid bacterium called *Lactiplantibacillus plantarum* to study whether and how this group of bacteria use extracellular electron transfer. Analysis of *L. plantarum* and its effect on its surroundings showed that these bacteria use a hybrid process to produce energy: the cells use aspects of extracellular respiration to increase the yield and efficiency of fermentation. Combining these two approaches may allow *L. plantarum* to adapt to different environments and grow faster, allowing it to compete against other species.

Tejedor-Sanz, Stevens et al. provide new information on a widespread group of bacteria that are often used in food production and industry. The next step will be to understand how the hybrid system is controlled and how it varies among species. Understanding this process could result in new biotechnologies and foods that are healthier, produce less waste, or have different tastes and textures.

All organisms possess mechanisms to conserve energy, that is, to convert light or chemical energy into cellular energy in the form of ATP (*Russell and Cook, 1995*). During respiration, microorganisms rely on either oxygen (aerobic respiration) or other exogenous substrates (anaerobic respiration) as terminal electron acceptors. Some microorganisms, most notably *Geobacter* spp., can anaerobically respire using electron acceptors outside the cell, such as iron (III) oxides or an electrode (*Renslow et al., 2013*; *Richter et al., 2012*). This process is called extracellular electron transfer (EET). Regardless of the identity of the electron acceptor, ATP synthesis during respiration occurs via oxidative phosphorylation (*Kim and Gadd, 2019*). In oxidative phosphorylation, electrons from electron carriers are transported by an electron transport chain, which creates a proton motive force (PMF) for ATP generation. Under anaerobic conditions, some cells can also conserve energy using fermentation. In fermentation, microorganisms use internally supplied electron acceptors, and ATP is generated mainly through substrate-level phosphorylation (*Kim and Gadd, 2019*). In substrate-level phosphorylation, ATP is generated in the cytoplasm by transfer of phosphate from metabolic intermediates to ADP (*Kim and Gadd, 2019*).

Lactic acid bacteria (LAB) are a diverse group of aerotolerant, saccharolytic microorganisms in the Firmicutes phylum which mainly use fermentation for energy conservation. LAB are essential for many food fermentations, including fermented milk and meats, fruits and vegetables, and grains (*Tamang et al., 2020*). Strains of LAB are also used for industrial chemical production (*Sauer et al., 2017*) and as probiotics to benefit human and animal health (*Vinderola et al., 2019*). LAB are generally grouped by their differences in hexose metabolism (*Salvetti et al., 2013*). Some species perform homofermentation, reducing pyruvate to lactate as the sole metabolic end-product from glycolysis. Other LAB perform heterofermentation, producing lactate along with ethanol, acetate, and $CO_2$ by the phosphoketolase pathway. However, for redox balancing, homofermentative LAB can also shift to a mixed acid fermentation and heterofermentative LAB use alternative electron acceptors, like fructose or citrate (*Hansen, 2018*). Although some LAB can respire in the presence of heme and menaquinone, those bacteria are unable to synthesize heme and many are also auxotrophic for menaquinone (*Pedersen et al., 2012*). Even those species capable of respiration still use fermentation metabolism as the primary mechanism to conserve energy (*Pedersen et al., 2012*). Therefore, LAB growth rates and cell yields are constrained by access to electron acceptors used to maintain intracellular redox balance during substrate-level phosphorylation.

The bioenergetics of anaerobic bacteria have been tightly linked to oxidative phosphorylation for anaerobic respiration and substrate-level phosphorylation for fermentation. However, experimental evidence shows a concurrent use of oxidative phosphorylation and substrate-level phosphorylation. For instance, some yeasts perform respiro-fermentation to enhance ATP production (*Pfeiffer and Morley, 2014*). Another example is the electron bifurcating mechanism used by some fermentative microorganisms such as *Clostridium* spp. (*Herrmann et al., 2008*; *Li et al., 2008*). Through that energy conservation strategy, cells can generate extra ATP through oxidative phosphorylation (*Buckel and Thauer, 2013*; *Müller et al., 2018*). Along with other examples that are not fully understood (*Hunt et al., 2010Kracke et al., 2018*), these observations suggest metabolisms that combine aspects of fermentation and respiration may exist.

We recently discovered that *Listeria monocytogenes,* a facultative anaerobic pathogen known to rely on respiratory metabolism, uses EET to reduce $Fe^{3+}$ or an anode through a flavin-based extracellular electron transfer pathway (*Light et al., 2018*). Use of this pathway allowed *L. monocytogenes* to maintain intracellular redox balance via NADH oxidation. This capacity was associated with the presence of a gene locus, called a flavin-based EET (FLEET) locus, that was identified in many Grampositive species in the Firmicutes phylum, including LAB. Studies in individual species of LAB such as *Lactococcus lactis* (*Freguia et al., 2009*; *Masuda et al., 2010*), *Enterococcus faecalis* (*Hederstedt et al., 2020*; *Keogh et al., 2018*), and *Lactiplantibacillus pentosus* (*Vilas Boas et al., 2015*) show that they can perform EET with an anode endogenously, that is without addition of molecules foreign to their native niches. These observations are quite surprising because endogenous EET has been mainly associated with respiratory organisms, even though some of these organisms also possess fermentative-type metabolism (*Glasser et al., 2014*). Those observations also raise the question of whether the FLEET locus is functional in LAB and what, if any, role it plays in energy conservation and metabolism.

Here, we explored EET across LAB and studied the implications of this trait at a metabolic and energetic level in *Lactiplantibacillus plantarum*, a homofermentative LAB capable of mixed acid fermentation and which can respire in the presence of exogenous heme and menaquinone. *L. plantarum* is of particular interest as it is a generalist LAB species found in insect, animal, and human digestive tracts and is essential for the production of many fermented foods (*Behera et al., 2018*; *Duar et al., 2017*). These findings have significance for the understanding of energy conservation strategies in primarily fermentative microorganisms and on lactic acid fermentations in food biotechnology.

## Results

### *L. plantarum* reduces extracellular electron acceptors

To determine whether *L. plantarum* can reduce extracellular electron acceptors, we first measured its ability to reduce insoluble ferrihydrite (iron (III) oxyhydroxide). Incubation of the model strain *L. plantarum* NCIMB8826 in the presence of ferrihydrite showed that this strain reduces $Fe^{3+}$ to $Fe^{2+}$ (*Figure 1A* and *Figure 1—figure supplement 1A*). Viable cells are required for iron reduction and this activity is dependent on the presence of exogenous quinone (DHNA, 1,4-dihydroxy-2-naphthoic acid) (*Figure 1A* and *Figure 1—figure supplement 1A-B*). The requirement for DHNA was hypothesized because DHNA is a precursor of demethylmenaquinone (DMK), a membrane electron shuttle utilized by *L. monocytogenes* for EET (*Light et al., 2018*), and *L. plantarum* lacks a complete DHNA biosynthetic pathway (*Brooijmans et al., 2009a*). For full activity, an electron donor (such as mannitol or glucose) was required to be present (*Figure 1A* and *Figure 1—figure supplement 1A*). Like *L. monocytogenes* (*Light et al., 2018*), the addition of riboflavin during the iron reduction assay also increased $Fe^{3+}$ reduction in a dose-dependent manner (*Figure 1—figure supplement 1C*). Thus, *L. plantarum* reduces insoluble iron in a manner similar to *L. monocytogenes*.

Next, we investigated whether the ability of *L. plantarum* to reduce insoluble iron was altered by growth media. *L. plantarum* was able to reduce iron after growth in either complete (MRS) medium or chemically defined medium (CDM) (*Figure 1—figure supplement 1B*). Iron reduction was greater when mannitol, a sugar alcohol, rather than glucose, was provided as the sole carbon source in MRS (*Figure 1—figure supplement 1B*). However, reduction was highest when *L. plantarum* was incubated in mannitol-containing MRS (mMRS) with both DHNA and ferric ammonium citrate present (*Figure 1—figure supplement 1D*). The addition of riboflavin to the growth medium did not further

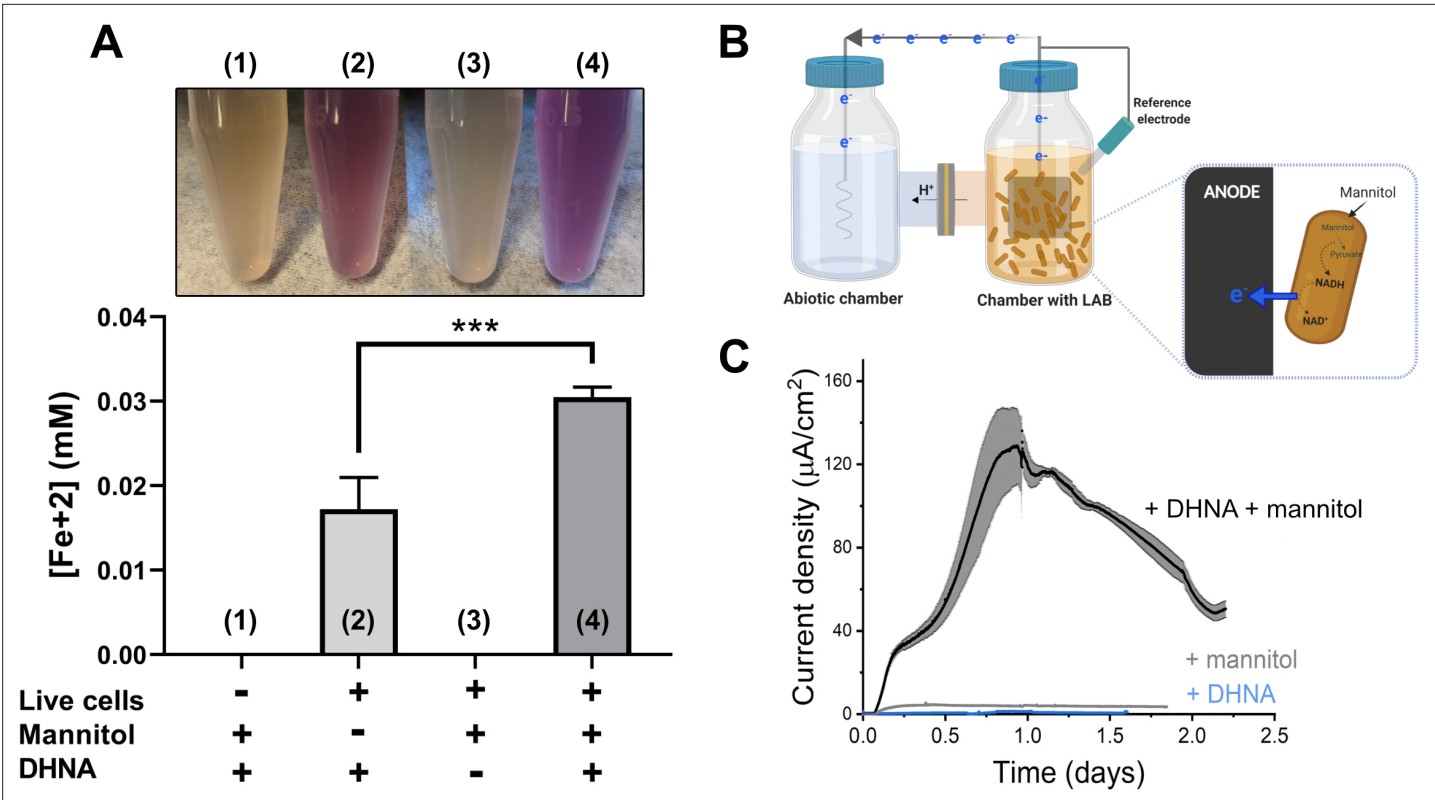

**Figure 1.** *L. plantarum* can reduce both $Fe^{3+}$ and an anode through EET. (**A**) Reduction of $Fe^{3+}$ (ferrihydrite) to $Fe^{2+}$ by *L. plantarum* NCIMB8826 after growth in mMRS. The assays were performed in PBS supplemented with 20 µg/mL DHNA and/or 55 mM mannitol. $Fe^{2+}$ was detected colorimetrically using 2 mM ferrozine. For *L. plantarum* inactivation, cells were incubated at 85°C in PBS for 30 min prior to the assay. Significant differences were determined by one-way ANOVA with Tukey's post-hoc test (n = 3), *** $p \leq 0.001$. (**B**) Two-chambered electrochemical cell setup for measuring current generated by *L. plantarum*. (**C**) Current density production over time by *L. plantarum* in CDM supplemented with 20 µg/mL DHNA and/or 110 mM mannitol. The anode was polarized at $+0.2V_{Ag/AgCl}$. The avg ± stdev of three biological replicates is shown. See also *Figure 1—figure supplement 1* and *Figure 1—figure supplement 2* and related data in *Figure 1—source data 1*.

The online version of this article includes the following source data and figure supplement(s) for figure 1:

**Source data 1.** Raw data of *Figure 1A and B*.

**Figure supplement 1.** Iron reduction by *L. plantarum* is dependent upon DHNA, carbon source, and riboflavin.

**Figure supplement 1—source data 1.** Raw data of *Figure 1—figure supplement 1A-F*.

**Figure supplement 2.** Current production by *L. plantarum* is a biotic process dependent on DHNA, carbon source, and riboflavin.

**Figure supplement 2—source data 1.** Raw data of *Figure 1—figure supplement 2A-D*.

**Figure supplement 3.** A sub-physiological concentration of DHNA stimulates EET in *L. plantarum*.

**Figure supplement 3—source data 1.** Raw data of *Figure 1—figure supplement 3A-C*.

increase iron reduction by *L. plantarum* (*Figure 1—figure supplement 1E*), potentially because riboflavin is already present in high quantities in MRS, a medium containing yeast extract (*Tomé, 2021*). Thus, *L. plantarum* was grown in mMRS supplemented with DHNA and iron before ferrihydrite reduction assays in all subsequent experiments.

*L. plantarum* EET activity was confirmed in a bioelectrochemical reactor by quantifying electron output as current (*Figure 1B*). *L. plantarum* reduced a carbon electrode (anode) polarized to +200 $mV_{Ag/AgCl}$ in the presence of both DHNA and an electron donor (mannitol) (*Figure 1C*). No current was observed in the absence of *L. plantarum* (*Figure 1—figure supplement 2A*), indicating that current production stems from a biological process. *L. plantarum* produced a maximum current of 129 ± 19 µA/cm² in mCDM (*Figure 1C*) and 225 ± 9 µA/cm² in mMRS (*Figure 1—figure supplement 2B*). Under EET conditions in mCDM, the *L. plantarum* biomass was 2.7 mg (dry cell mass). Assuming 50% of the dry cell mass was protein, the specific electron transfer rate was 57 µmol electrons/mg- protein/

hr and the current production was 1.5 mA/mg-protein. This value is lower than that reported for *Geobacter sulfurreducens* (4–8 mA/mg-protein) (*Marsili et al., 2010*; *Rose and Regan, 2015*), the model species for direct EET, and higher than that of *Shewanella oneidensis* (0.67 mA/mg-protein) (*Marsili et al., 2008*), the model species for mediated EET. It should be noted that these species, unlike *L. plantarum*, can synthesize riboflavin and quinones and do not require the addition of either for EET activity. Similar to our iron reduction experiments, EET to an anode occurred with different electron donors and growth media (*Figure 1—figure supplement 2B-C*), and current increased after supplementation of riboflavin when it was omitted from the growth medium (*Figure 1—figure supplement 2D*). Because of these differences, CDM was amended with mannitol and riboflavin in subsequent experiments.

DHNA is found in concentrations of 0.089–0.44 µg/mL in commercial fermented beverages (*Eom et al., 2012*), and under laboratory conditions, microbes can synthesize and secrete DHNA leading to concentrations of 0.37–48 µg/mL (*Isawa et al., 2002*; *Furuichi et al., 2006*; *Kang et al., 2015*). To test whether EET in *L. plantarum* is relevant under these physiological concentrations, we probed whether *L. plantarum* can perform EET with a sub-physiological DHNA concentration of 0.01 µg/mL. Indeed, *L. plantarum* can reduce iron and produce significant current density (*Figure 1—figure supplement 3A-B*), although the magnitude of iron reduction and current was smaller than what was observed with 20 µg/mL. These results show that the concentrations of DHNA found in niches of *L. plantarum* can support EET and suggest the magnitude of EET will depend on the DHNA concentration.

## Iron reduction by LAB is associated with the presence of *ndh2* and *pplA*

Because iron reduction by *L. monocytogenes* requires the genes in a 8.5 kb gene locus encoding a flavin-based EET (FLEET) pathway (*Light et al., 2018*), we looked for the presence of these genes in 1,788 LAB genomes deposited in NCBI. Homology searches identified the complete FLEET locus in 11 out of 38 genera including diverse LAB such as *Enterococcus* and *Lacticaseibacillus* (*Figure 2A*). The other LAB genera either lack multiple FLEET pathway genes or, as was observed for all 68 strains of *Lactococcus*, contain all genes except for *pplA*, which is predicted to encode an extracellular flavin-binding reductase. Among the lactobacilli, genomes of 19 out of 94 species contain the entire FLEET system (*Figure 2—figure supplement 1*). The lactobacilli species with the entire FLEET locus are homofermentative and are distributed between different phylogenetic groups (*Zheng et al., 2020*). These data show that the FLEET locus is conserved across LAB genera besides *L. plantarum*, including other homofermentative LAB species known to colonize host and food environments.

To determine whether LAB FLEET gene presence was associated with EET activity, a diverse collection of LAB strains were examined for their capacity to reduce ferrihydrite. The assay showed that isolates of *L. plantarum*, *Lactiplantibacillus pentosus*, *Lacticaseibacillus rhamnosus*, *Lacticaseibacillus casei*, *Enterococcus faecium*, and *Enterococcus faecalis* are capable of $Fe^{3+}$ reduction (*Figure 2B*). The genomes of those species also contain a complete FLEET locus (*Figure 2A* and *Figure 2—figure supplement 1*). Conversely, strains of *Lactococcus lactis*, *Ligilactobacillus murinus*, *Levilactobacillus brevis*, *Pediococcus pentosaceus*, and *Streptococcus agalactiae* showed little to no iron reduction activity (*Figure 2B*). The presence of FLEET-associated genes varied between those species, but only strains of species found to contain both *ndh2*, a predicted membrane-bound, type-II NADH dehydrogenase, and *pplA* were able to reduce iron under the conditions tested.

*L. plantarum* NCIMB8826 exhibited the highest EET activity resulting in at least 2.5-fold greater $Fe^{3+}$ reduction than the other *L. plantarum* strains (*Figure 2B*). Remarkably, however, the *L. plantarum* NCIMB8826 genome and the genomes of 138 other *L. plantarum* strains queried all harbored a complete FLEET locus including *ndh2* and *pplA* (*Figure 2—figure supplement 1* and *Figure 2—figure supplement 2A*). Among those strains tested for the capacity to reduce $Fe^{3+}$, *L. plantarum* NCIMB700965 and 8.1 could not reduce $Fe^{3+}$ but possessed all genes in the FLEET pathway. Closer examination of both strains by aligning their FLEET loci with NCIMB8826 revealed unique IS30-family transposons in the intergenic promoter regions spanning *ndh2* and *pplA* (*Figure 2—figure supplement 2A*). These genes were minimally expressed in *L. plantarum* NCIMB700965 and 8.1 in comparison to NCIMB8826 (*Figure 2—figure supplement 2B*). *ndh2* and *pplA* were also the only two genes in the FLEET gene locus that were induced when *L. plantarum* NCIMB8826 was incubated in mMRS supplemented with DHNA and iron (*Figure 2C* and *Figure 1—figure supplement 1D*). Both *ndh2*

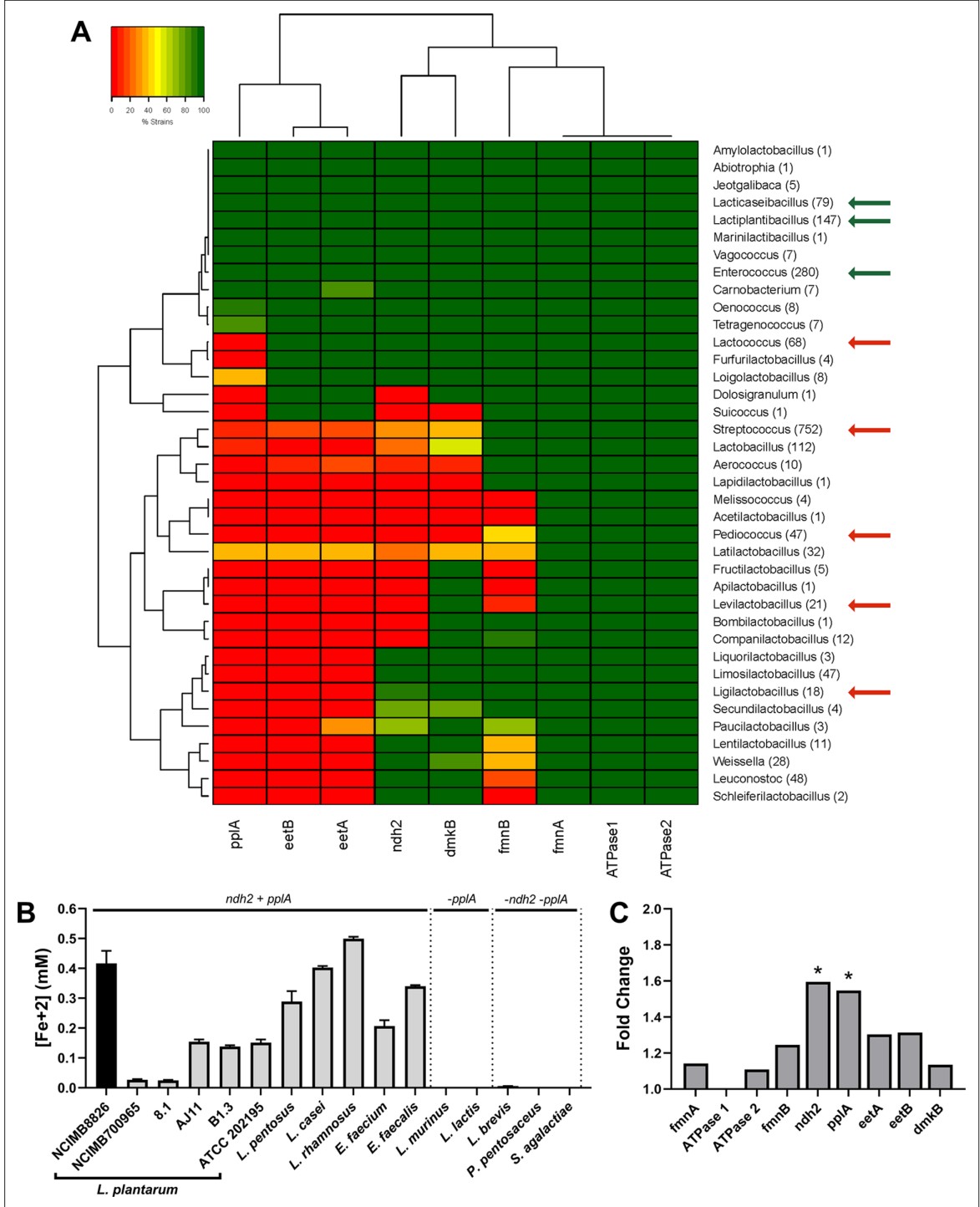

**Figure 2.** The FLEET genes *ndh2* and *pplA* are associated with iron reduction by LAB. (**A**) Heatmap showing the genera in the Lactobacillales order containing FLEET genes. Homology searches were conducted using tBLASTx for 1788 complete LAB genomes in NCBI (downloaded 02/25/2021) against the *L. plantarum* NCIMB8826 FLEET locus. A match was considered positive with a Bit-score >50 and an E-value of <10⁻³. Arrows designate genera tested for iron reduction activity; green = EET-active with $Fe^{3+}$, red = EET-inactive with $Fe^{3+}$. (**B**) Reduction of ferrihydrite in PBS with 20 µg/mL DHNA and 55 mM mannitol after growth in mMRS supplemented with 20 µg/mL DHNA and 1.25 mM ferric ammonium citrate. The avg ± stdev of three biological replicates per strain is shown. (**C**) Relative expression of NCIMB8826 FLEET locus genes in mMRS with 20 µg/mL DHNA and 1.25 mM ferric ammonium citrate compared to growth in mMRS. Significant differences in expression were determined by the Wald test (n = 3) with a $Log_2$ (fold change) > 0.5 and an FDR-adjusted p-value of <0.05. See also *Figure 2—figure supplement 1* and *Figure 2—figure supplement 2* and related data in *Figure 2—source data 1*.

*Figure 2 continued on next page*

*Figure 2 continued*

The online version of this article includes the following source data and figure supplement(s) for figure 2:

**Source data 1.** EET locus homology matches at genus level via tBLASTx from complete Lactobacillales genome dataset for *Figure 2A*, raw data for *Figure 2B* and relative expression data of *Figure 2C*.

**Figure supplement 1.** Conservation of FLEET locus genes among lactobacilli.

**Figure supplement 1—source data 1.** EET locus homology dataset of *Figure 2—figure supplement 1*.

**Figure supplement 2.** *ndh2* and *pplA* are required for iron reduction through EET.

**Figure supplement 2—source data 1.** Relative expression dataset of *Figure 2—figure supplement 2*.

and *pplA* were induced (~1.6 fold, $p \leq 0.05$) in MRS containing mannitol, DHNA, and ferric ammonium citrate (*Figure 2C*), but were not upregulated when either DHNA or ferric ammonium citrate were omitted from the culture medium (*Figure 2—figure supplement 2C*). Taken together, these data show that widespread iron reduction in LAB is tightly associated with the presence and upregulation of *ndh2* and *pplA*, suggesting they are required for EET.

## Ndh2 is required and PplA is conditionally required for *L. plantarum* EET

In order to confirm the necessity of *ndh2* and *pplA* for EET in *L. plantarum*, we constructed *ndh2* and *ppA* deletion mutants of *L. plantarum* NCIMB8826. Both mutants were significantly impaired in their capacities to reduce ferrihydrite compared with the wild-type strain (*Figure 3A*). The *ndh2* and *pplA* deletion mutants also had different effects on the oxidation-reduction potential (ORP) of mMRS. ORP is defined as the ratio of all oxidative to reductive components in an environment (*Killeen et al., 2018*) and is an important environmental condition which influences the outcome of LAB fermentations such as flavor development in cheese (*Morandi et al., 2016*) and the growth of spoilage microorganisms (*Olsen and Pérez-Díaz, 2009*). Expectedly for the *L. plantarum* EET phenotype, significant reductions in mMRS ORP only occurred during *L. plantarum* growth when DHNA was included in the culture medium (*Figure 3—figure supplement 1A*). Although ORP declined for all three strains in a manner consistent with other ORP-reducing enzymatic activities (for example the reduction of oxygen by NADH oxidase) (*Tachon et al., 2010*), wild-type *L. plantarum* resulted in greater reductions in ORP compared to either mutant in mMRS, and these differences were significant at most time points measured over a 12 hr period (*Figure 3B*). The effects on ORP occurred in the absence of changes in growth rates, cell yields, and medium pH (*Figure 3—figure supplement 1A-D*). The $\Delta mV_{max}$ was reached during mid-exponential phase (approximately 5 hr) (*Figure 3—figure supplement 1B*), and at that time, wild-type *L. plantarum* cells but not the $\Delta ndh2$ or $\Delta pplA$ strains were active in the ferri-hydrite reduction assay (*Figure 3—figure supplement 1E*). This difference in ferrihydrite reduction activity similarly persisted in stationary phase cells (12 hr) (*Figure 3—figure supplement 1F*). These observations show that *ndh2* and *pplA* contribute to the capacity of *L. plantarum* to reduce iron and have relevance to the ORP-dependent activities occurring during food fermentations (*van Dijk et al., 2000*).

Use of an anode as an external electron acceptor instead of ferrihydrite showed a similar, but not identical genetic dependency. *L. plantarum* $\Delta ndh2$ produced a significantly lower current density (*Figure 3C*) and a lower peak current (*Figure 3—figure supplement 2A*). Surprisingly, *L. plantarum* $\Delta pplA$ was able to produce the same amount of current as the wild-type strain, suggesting that the lipoprotein PplA is not essential and might not be involved in anode reduction through EET. This observation led us to investigate the anodic-EET ability of other LAB species lacking *pplA* like *Lactococcus lactis* (*Figure 3D*). DHNA was not provided to these strains because they can synthesize demethyl-menaquinone and other quinones (*Rezaïki et al., 2008*). Both *L. lactis* strain IL1403 and strain KF147 were capable of current generation, confirming that PplA is not essential for LAB to produce current. This is consistent with the finding that other extracellular reductases besides PplA are responsible for EET activity in Gram-positive bacteria (*Light et al., 2019*). Taken together these results show that EET activity is dependent upon the presence of the putative FLEET locus, and specifically *ndh2* and conditionally *pplA*.

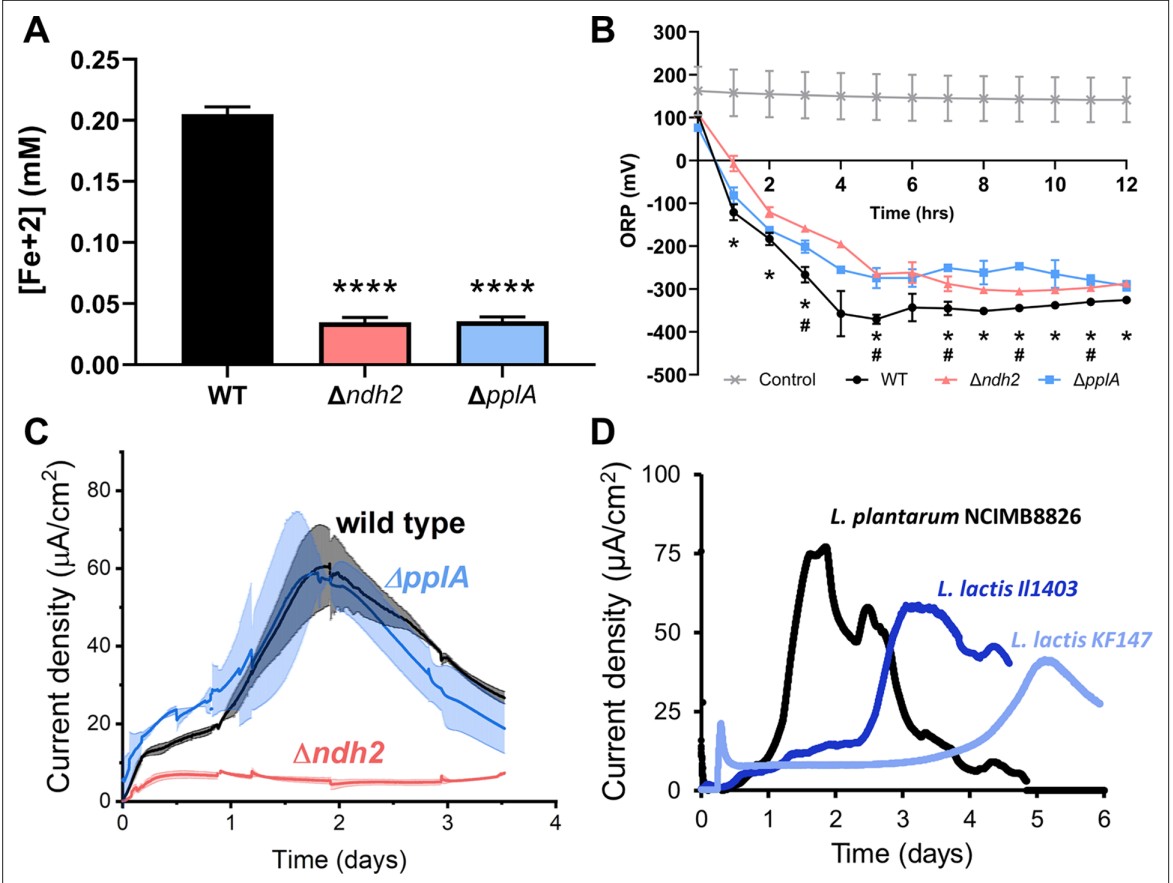

**Figure 3.** *L. plantarum* requires *ndh2* and conditionally *pplA* for EET. (**A**) Reduction of $Fe^{3+}$ (ferrihydrite) to $Fe^{2+}$ with wild-type *L. plantarum* or EET deletion mutants in the presence of 20 µg/mL DHNA and 55 mM mannitol after growth in mMRS supplemented with 20 µg/mL DHNA and 1.25 mM ferric ammonium citrate. Significant differences determined by one-way ANOVA with Dunnett's post-hoc test, **** $p \leq 0.0001$. (**B**) Redox potential of mMRS supplemented with 20 µg/mL DHNA and 1.25 mM ferric ammonium citrate after inoculation with wild-type *L. plantarum* or EET deletion mutants. Significant ORP differences between the wild-type and mutant strains determined by two-way RM ANOVA with Tukey's post-hoc test, * $p < 0.05$ (WT vs. Δ*ndh2*); # $p < 0.05$ (WT vs. Δ*pplA*). (**C**) Current density generated by wild-type *L. plantarum* and deletion mutants in mCDM supplemented with 20 µg/mL DHNA. The avg ± stdev is shown. (**D**) Current density generated by *L. plantarum* and two *L. lactis* strains lacking *pplA* in mCDM. For *L. plantarum*, the mCDM was supplemented with 20 µg/mL DHNA. The data correspond to the average of two (**D**) or three (**A** to **C**) biological replicates per strain. See also *Figure 3—figure supplement 1* and *Figure 3—figure supplement 2* and related data in *Figure 3—source data 1*.

The online version of this article includes the following source data and figure supplement(s) for figure 3:

**Source data 1.** Raw data of *Figure 3A–C*.

**Figure supplement 1.** Impact of *ndh2* and *pplA* deletion on growth, iron reduction, current density, and metabolites production.

**Figure supplement 1—source data 1.** Raw data of *Figure 3—figure supplement 1A-F*.

**Figure supplement 2.** Impact of *ndh2* and *pplA* deletion on maximum current density, pH and metabolites production.

**Figure supplement 2—source data 1.** Raw data of *Figure 3—figure supplement 2A-C*.

## *L. plantarum* increases energy conservation and balances intracellular redox state when performing EET

Building from studies in *E. faecalis* (*Keogh et al., 2018*), it has been suggested that EET improves growth by either enabling iron to be acquired as a macronutrient or by enhancing respiration (*Jeuken et al., 2020*). It is worth noting that several studies have shown that *L. plantarum* does not require iron to grow (*Elli et al., 2000*; *Weinberg, 1997*). To test whether EET allowed increased iron acquisition by *L. plantarum*, we measured intracellular iron by Inductively Coupled Plasma-Mass Spectrometry (ICP-MS). There was no significant difference in intracellular iron concentrations between *L. plantarum* growth in mMRS supplemented with DHNA and iron compared to growth in mMRS alone (*Figure 4—figure supplement 1*). Moreover, deletion of *ndh2* did not significantly change the amount

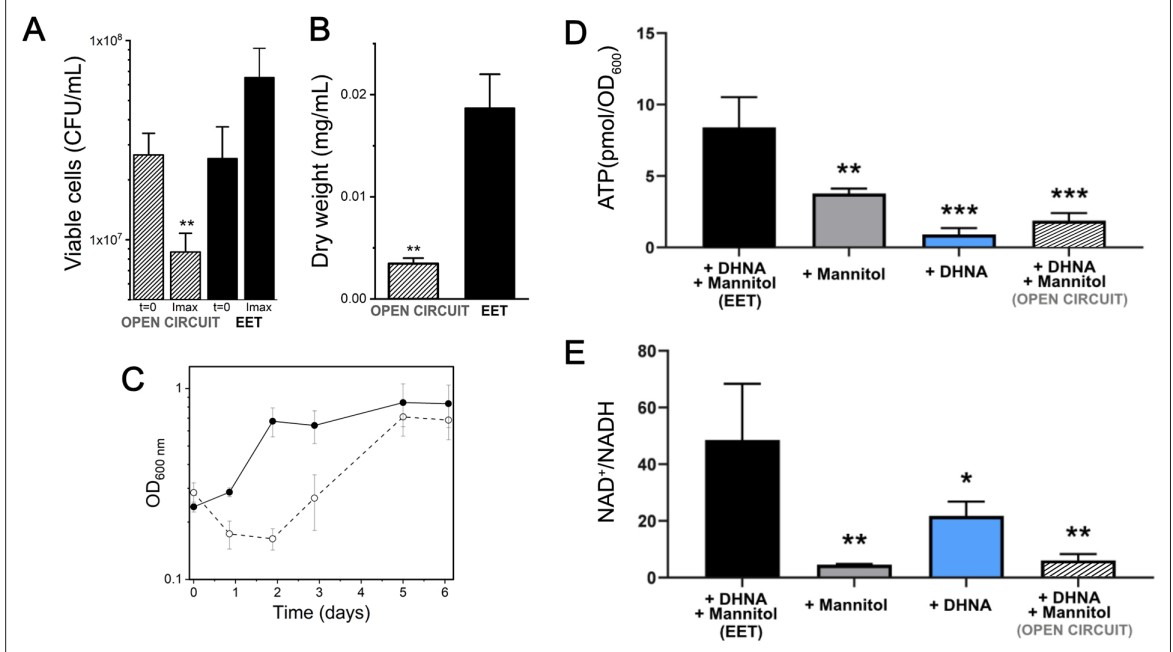

**Figure 4.** Growth, ATP, and redox balance of *L. plantarum* changes when an anode is provided as an extracellular electron acceptor. These measurements and the current density plot shown in *Figure 1C* are from the same experiment. (**A**) Viable cells and (**B**) dry weight at the point of maximum current density under current circulating conditions (EET) and at open circuit conditions (OC) at the same time point. (**C**) Change in cell numbers measured by $OD_{600}$ over time in the bioreactors under EET (continuous line) and OC conditions (dotted lines). (**D**) ATP production per $OD_{600}$ unit and (**E**) $NAD^+/NADH$ ratios at the point of maximum current density. The bioreactors were shaken vigorously to dislodge cells before sampling. The avg ± stdev of three biological replicates is shown. Significant differences were determined by one-way ANOVA with (**A and B**) Dunn-Sidak post-hoc test (n = 3) and (**D and E**) Dunnett's post-hoc test (n = 3), * $p < 0.05$; ** $p < 0.01$; *** $p < 0.001$; **** $p < 0.0001$. See also *Figure 1* panel C and *Figure 4—figure supplement 3* and related data in *Figure 4—source data 1*.

The online version of this article includes the following source data and figure supplement(s) for figure 4:

**Source data 1.** Raw data of *Figure 4A–D*.

**Figure supplement 1.** Intracellular metal concentrations in *L. plantarum* are not affected by EET-conducive growth conditions.

**Figure supplement 1—source data 1.** Raw data of *Figure 4—figure supplement 1*.

**Figure supplement 2.** Redox-active metal concentrations in *L. plantarum* are not affected by the presence of *ndh2*.

**Figure supplement 2—source data 1.** Raw data of *Figure 4—figure supplement 1*.

**Figure supplement 3.** Use of $Fe^{3+}$ as an electron acceptor allows *L. plantarum* to regenerate $NAD^+$.

**Figure supplement 3—source data 1.** Raw data of *Figure 4—figure supplement 3*.

of intracellular iron (*Figure 4—figure supplement 1*). ICP-MS showed that other redox-active metals used for EET, such as manganese and copper (*Kouzuma et al., 2012*; *Fan et al., 2018*) were also not affected (*Figure 4—figure supplements 1 and 2*). In contrast to studies in *E. faecalis* in which iron supplementation leads to intracellular accumulation of this metal (*Keogh et al., 2018*), these data show that *L. plantarum* does not use EET to increase its acquisition of iron or other redox-active metals, suggesting EET may instead play a role in energy conservation.

We next sought to understand if EET impacts energy conservation in *L. plantarum* by comparing its growth and ATP levels in the presence of a polarized anode. The highest current density (i.e. greatest EET activity) produced by *L. plantarum* in mannitol CDM typically occurred within 24 hr after inoculation into the bioreactor (*Figure 1C*). At this point, there was an approximately 4-fold higher dry cell weight and 2-fold higher numbers of viable cells compared to *L. plantarum* incubated in open circuit (OC) conditions (*Figure 4A–B*) Current density declined from its maximum value when *L. plantarum* cells performing EET were in exponential growth (*Figures 1C and 4C*). By comparison, growth was not observed until two days later under OC conditions (*Figure 4C*). During peak current production, intracellular ATP levels were significantly higher (4.5-fold) under EET compared to OC conditions (*Figure 4D* and *Table 1*). These results strongly suggest faster ATP accumulation under EET conditions

**Table 1.** Bioenergetic balances suggest energy conservation under EET conditions occurs via substrate-level phosphorylation.

The reactors contained 20 µg/mL of DHNA and mannitol as the electron donor. Balances were calculated with data obtained by day four from *Figure 5*. See also *Supplementary file 1*. SLP stands for substrate-level phosphorylation. $Y_{fermentation}$ refers to the total fermentation products obtained (see *Supplementary file 1*) per mol of sugar consumed. $Y_{mannitol}$ is the ATP produced from the total fermentation products per mol of sugar consumed, and $Y_{ATP}$ is the dry weight measured per mol of ATP produced from fermentation products.

| | NADH consumed* | Calculated ATP† (from metabolites) | Biomass yield | $Y_{fermentation}$ | $Y_{mannitol}$ | $Y_{ATP}$ |
|---|---|---|---|---|---|---|
| Units | mM | mM | g-dw/mol-mannitol | mmol product/ mmol-mannitol | mol ATP/mol mannitol | g dw/mol ATP |
| EET | 6.44 ± 0.48 via anode 16.69 ± 2.72 via SLP | 16.6 ± 1.5 | 4.85 ± 0.33 | 1.53 ± 0.13 | 1.59 ± 0.13 | 3.09 ± 0.36 |
| OC | 5.51 ± 0.97 via SLP | 5.7 ± 0.6 | 7.21 ± 1.41 | 0.87 ± 0.09 | 0.89 ± 0.09 | 8.06 ± 0.86 |

*Calculated based on production of 3 mol of NADH produced per mol of mannitol, 1 mol of $NAD^+$ per lactate, 2 mol of $NAD^+$ per ethanol, 2 mol of $NAD^+$ mol succinate produced and 0.5 mmol of $NAD^+$ per mol of electrons harvested on the anode.
†Calculated based on production of 1 mol of ATP per lactate, 2 mol per acetate, 1 mol per ethanol, and 3 mol per succinate produced.

allowed *L. plantarum* to exit lag phase more rapidly. ATP levels were also greater in *L. plantarum* when in the presence of both mannitol and DHNA, compared to either mannitol or DHNA separately (*Figure 4D*). Thus, EET allows *L. plantarum* to initiate growth and accumulate ATP more rapidly, indicating that EET significantly increases energy conservation in *L. plantarum*.

Because fermentation, anaerobic respiration, and aerobic respiration are each associated with a different $NAD^+$/NADH ratio, energy conservation is linked to intracellular redox homeostasis (*Holm et al., 2010*). Therefore, we probed redox homeostasis in *L. plantarum* under EET conditions by measuring intracellular $NAD^+$/NADH at the point of maximum current density (*Figure 4E*). *L. plantarum* showed an 8-fold higher $NAD^+$/NADH ratio under EET conditions compared to OC (*Figure 4E*). This result was not limited to the presence of a polarized anode as *L. plantarum* also contained a significantly higher $NAD^+$/NADH ratio when $Fe^{3+}$ was available as a terminal electron acceptor (*Figure 4—figure supplement 3*). These $NAD^+$/NADH ratios are more similar to those found for in *E. coli* performing aerobic respiration (*de Graef et al., 1999*) or *G. sulfurreducens* performing anaerobic respiration than in LAB performing fermentation (*Guo et al., 2017*). Taken together, our data indicate that EET is involved in energy conservation, and the intracellular redox balance during EET mimics a respiratory rather than a fermentative process.

## EET increases fermentative metabolism through substrate-level phosphorylation and reduction in extracellular pH

Metal-reducing bacteria use EET in anaerobic respiration (*Richter et al., 2012*; *Shi et al., 2007*). Ndh2 is considered an anaerobic respiratory protein, and *L. plantarum* can perform anaerobic respiration with exogenous menaquinone and heme using an electron transport chain (*Brooijmans et al., 2009b*). This led us to hypothesize that those electron transport proteins could also be involved for EET to conserve energy as part of anaerobic respiration. To test this hypothesis, we examined whether any of the known electron transfer proteins needed for PMF generation in aerobic and anaerobic respiration are required for *L. plantarum* EET. Neither the addition of heme to restore bd-type cytochrome (*cydABCD*) used in aerobic respiration, nor deletion of the respiratory nitrate reductase (Δ*narGHJI*) significantly altered current production (*Figure 5—figure supplement 1A-B*). Because Ndh2 is a type-II NADH dehydrogenase which does not contribute to a proton gradient (*Lin et al., 2008*; *Nakatani et al., 2020*), these observations show that while EET does involve a respiratory protein, it does not involve any of the known PMF-generating electron transfer proteins in *L. plantarum*.

Respiration is also associated with the tricarboxylic acid (TCA) cycle. *L. plantarum*, like other LAB, does not possess an oxidative branch of the TCA cycle and only contains a reductive branch (*Tsuji et al., 2013*). To probe whether the reductive branch was active during EET, we also examined production of succinate, the terminal end-product of the reductive branch. EET did not increase the succinate

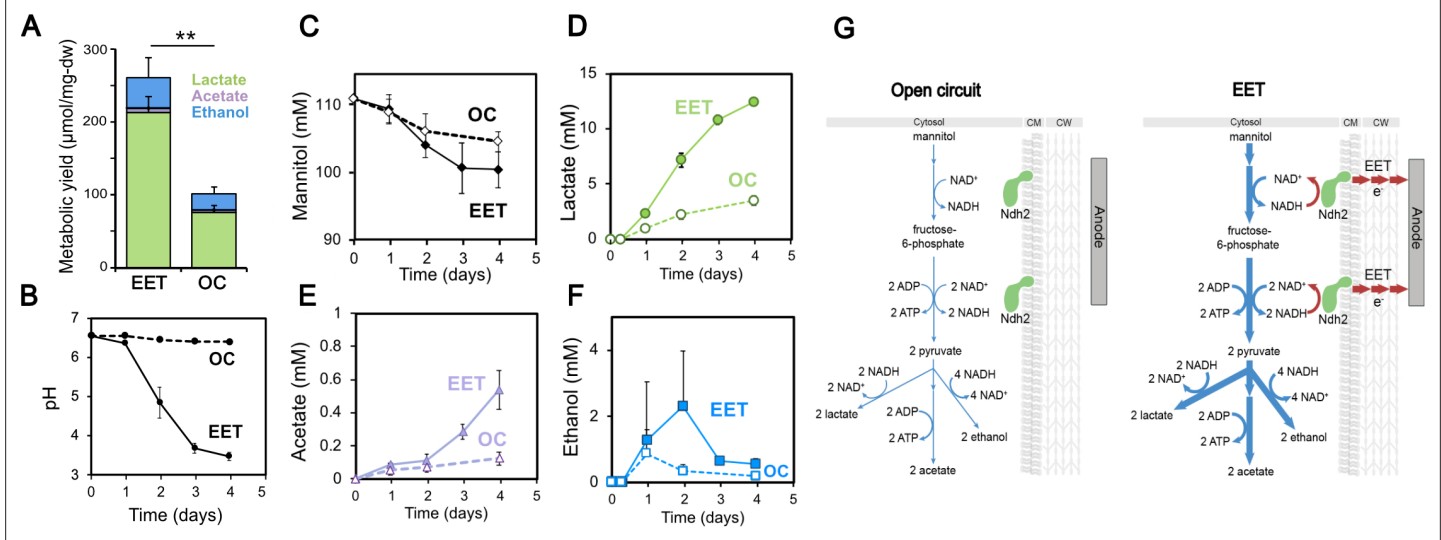

**Figure 5.** Fermentation fluxes are increased when an anode is provided as an extracellular electron acceptor. Results are from the same set of experiments as the current density plot shown in *Figure 3C*. (**A**) Metabolic yields of *L. plantarum* end-fermentation products under open circuit conditions (OC) and current circulating conditions (EET) in mCDM supplemented with 20 µg/mL DHNA. (**B**) pH measurements and (**C**) mannitol, (**D**) lactate, (**E**) acetate, and (**F**) ethanol concentrations over time under OC and EET conditions. (**G**) Schematic of proposed model for NADH regeneration during fermentation of mannitol in the presence of an anode as electron sink for *L. plantarum*. The avg ± stdev of three biological replicates is shown. Significant differences were determined by one-way ANOVA with Dunn-Sidak post-hoc (n = 3), ** $p \leq 0.01$. See also *Figure 5—figure supplement 1* and *Figure 5—figure supplement 2* and related data in *Figure 5—source data 1*.

The online version of this article includes the following source data and figure supplement(s) for figure 5:

**Source data 1.** Raw data of *Figure 5A–F*.

**Figure supplement 1.** EET by *L. plantarum* is not dependent on aerobic or anaerobic respiration components.

**Figure supplement 1—source data 1.** Raw data of *Figure 5—figure supplement 1A, B*.

**Figure supplement 2.** EET by *L. plantarum* does not involve TCA cycle metabolites.

**Figure supplement 2—source data 1.** Raw data of *Figure 5—figure supplement 1*.

concentration (*Figure 5—figure supplement 2*). Moreover, we did not detect any intermediates of the reductive branch of the TCA cycle, that is oxalacetate, malate, or fumarate. This indicates that EET did not cause additional metabolic flux through its TCA cycle. Thus, none of the known metabolic pathways or electron transport proteins associated with anaerobic respiration, besides Ndh2, are required for EET. These results suggest increased energy conservation during EET in *L. plantarum* is not through canonical anaerobic respiration.

An alternative hypothesis is that increased energy conservation under EET conditions is driven by changes in fermentation. *L. plantarum* uses glycolysis to convert mannitol to two molecules of pyruvate which are then converted mainly to lactate or ethanol via NADH-consuming steps, or acetate via an ATP-generating reaction using substrate-level phosphorylation (*Dirar and Collins, 1972*). Thus, shifting toward production of acetate from to lactate or ethanol production can increase ATP yield during fermentation. Additionally, NADH can be re-generated by oxidizing pyruvate to yield 2,3-butanediol, using acetoin as an intermediate. Fermentation in *L. plantarum* also decreases the pH of the surrounding media.

To probe changes in fermentation, we measured the concentrations of mannitol, acetate, lactate, ethanol, acetoin, 2,3-butanediol, formate, and pyruvate and the pH in *L. plantarum* cultures during OC and EET conditions. After four days, we accounted for ~80% and ~ 55% of the total carbon under EET and OC conditions (for all metabolite concentrations see *Supplementary file 1*), giving us a quantitative view of metabolism under EET conditions. Surprisingly, under EET conditions, the distribution of major end-fermentation products (acetate, lactate, and ethanol) did not change, but their yield per cell was 2.6-fold higher compared to OC conditions (*Figure 5A*). While we did not detect acetoin or 2,3-butanediol, formate was found at trace levels, and pyruvate was found at similar, low levels under EET and OC conditions (*Figure 5—figure supplement 2*). After accounting for mannitol

consumption, we observed that EET allowed cells to produce ~1.75 x more fermentation products per each mol of mannitol utilized ($Y_{fermentation}$, *Table 1*). The culture medium pH was also significantly lower than under OC (*Figure 5B*), a result which may indicate that EET conferred higher levels of acid stress on *L. plantarum*, and therefore, reductions in cell viability, despite EET leading to higher cell numbers overall (as measured by dry cell weight) (*Figure 4A–B*). A similar acidification of the medium was observed for Δ*pplA*, but not for Δ*ndh2*, when an anode was present as electron acceptor, indicating that *ndh2*-dependent EET is needed to decrease the pH (*Figure 3—figure supplement 2B*). When much lower, sub-physiological levels of DHNA were supplied (0.01 µg/mL), a smaller but significant decrease in the pH of the medium was also observed (*Figure 1—figure supplement 3C*). Overall, these results show that EET allows *L. plantarum* to ferment to ~1.75 x greater extent and to acidify the medium to a greater extent as well.

We also observed that EET led to higher cellular metabolic fluxes, that is, higher changes in metabolites per cell per unit time. Although the final $OD_{600nm}$ and dry cell weight were not significantly different (*Supplementary file 1*), *L. plantarum* utilized mannitol and produced acetate and lactate more rapidly under EET than OC conditions (*Figure 5C–F*). Cells performing EET were ~2 fold faster at consuming mannitol (*Figure 5C*) between days 1 and 3. Mannitol consumption increased between day 1 and day 2, approximately when the cells transitioned to higher current density (*Figure 3C*), suggesting that increased EET drove that increased consumption. The overall rates of acetate and lactate production also increased 3.4 and 3.6 times (*Figure 5D–E*), respectively. Measurements of metabolites produced by Δ*pplA* and Δ*ndh2* strains confirmed that, like for current production to an anode, the EET-associated increased metabolic flux in *L. plantarum* requires the presence of Ndh2, but not PplA (*Figure 3—figure supplement 2C*). Overall, these data indicate that *ndh2*-dependent EET increases both the flux and final yield of fermentation in *L. plantarum*.

Because the production of acetate yields ATP, these results also suggested that the increase in ATP generation under EET conditions may be due to substrate-level phosphorylation. To probe whether EET-associated increase in fermentative flux could account for the changes in ATP generation, we calculated fermentation balances (*Table 1*). Our measurements account for 80% of the carbon under EET conditions (see *Supplementary file 1*), leaving a maximum of ~20% systematic uncertainty in these calculations. The concentrations of fermentation products detected (*Supplementary file 1*) were used to estimate the total ATP in the presence and absence of EET. The estimated ATP was 3-fold higher under EET conditions than OC conditions (*Table 1*), a result that is consistent with the ~2.5 fold higher accumulation of ATP measured at maximum current density (*Figure 4D*). Overall, this quantitative analysis shows that the vast majority of the increased energy conservation under EET conditions can be accounted for by an increase in fermentation yield and substrate-level phosphorylation.

## EET shifts how *L. plantarum* uses electron acceptors and converts ATP into biomass

Thus far, our results provided an unusual picture of the energy metabolism of *L. plantarum* under EET conditions; while EET significantly shifted the intracellular redox state to a more respiratory-like balance, its increased ATP yield was mainly accounted for by an increased fermentative yield. Another major difference in fermentation and anaerobic respiration is the use of the endogenous versus exogenous electron acceptors. To more deeply understand how *L. plantarum* uses organic molecules and the anode as electron acceptors when performing EET, the electron balances under EET and OC conditions were calculated (*Table 1* and *Supplementary file 1*). We estimated the NADH produced using two different methods (see *Supplementary file 1* for methodology) and the NADH re-oxidized through the reduction of the anode (measured as current) and via substrate-level phosphorylation. This allowed us to obtain a global balance of the $NAD^+/NADH$ ratio. Under OC conditions between 35% and 66% of the NADH produced from the oxidation of mannitol to pyruvate (a range is given using the two methods used) was re-oxidized to $NAD^+$ (5.5 mM NADH consumed, *Table 1*), qualitatively agreeing with the low $NAD^+/NADH$ ratios measured (*Figure 4E*). In contrast, electron balance calculations showed that between 77% and 96% of the NADH produced under EET conditions was re-oxidized (17 mM NADH consumed, *Table 1*), a result that is consistent with the significantly higher $NAD^+/NADH$ ratios measured (*Figure 4E*). Interestingly, these calculations estimate that 55–69% of the total NADH generated was oxidized through fermentation, while 21–28% of the NADH was oxidized using the electrode as a terminal electron acceptor (*Table 1*). Thus, *L. plantarum* growing

under EET conditions achieves a more oxidized intracellular redox balance by more completely fermenting mannitol to lactate and ethanol and by using the electrode as a terminal electron acceptor (*Figure 5G*). These observations reinforce that the energy metabolism of *L. plantarum* under EET conditions utilizes elements of both fermentation and anaerobic respiration.

In rapidly dividing cells, energy conservation, a catabolic process, is associated with growth, an anabolic process. However, catabolism need not be coupled with anabolism (*Russell and Cook, 1995*). To determine how catabolic and anabolism are connected under EET conditions, the ATP requirements to grow biomass ($Y_{ATP}$) were estimated using the calculated ATP and the measured dry weight. Under OC conditions, the $Y_{ATP}$ obtained (8.06 ± 0.8 g dw/mol ATP) for *L. plantarum* was similar to that observed previously (10.9 g-dw/mol ATP) (*Dirar and Collins, 1972*). Hence, without EET, the ATP generated from fermentation was converted into biomass nearly at the expected efficiency. In contrast, a significantly lower $Y_{ATP}$ was reached for *L. plantarum* performing EET (3.07 ± 0.35 g dw/mol ATP) (*Table 1*). This observation indicates that under EET conditions, either more ATP is required to produce biomass or more ATP is utilized by other functions such as for PMF-generation and intracellular pH regulation (*Russell and Cook, 1995*). EET conditions also resulted in 79% more ATP per mol of fermented mannitol ($Y_{mannitol}$). Consequently, molar biomass yields (g-dw/mol-mannitol) under EET conditions were significantly lower (*Table 1*), in agreement with previous observations in respiratory electroactive species (*Esteve-Núñez et al., 2005*). These calculations show that when *L. plantarum* performs EET, anabolism and catabolism processes are differently coupled than under OC conditions. ATP is produced more efficiently, but this it is less efficiently utilized to make biomass. Overall, these results show an intriguing pattern of coupling between anabolism and catabolism, indicative of a novel energy metabolism in *L. plantarum* during EET.

## EET is active in vegetable fermentations

Our results inspired us to explore whether EET could occur in a physiological niche of LAB such as fermented foods. LAB are necessary for the making of many fermented fruit and vegetable foods and the properties of those foods depend on the metabolic diversity of the LAB strains present (*Gänzle, 2015*). Plant tissues also contain a much wider variety of carbon substrates and potential electron acceptors than the CDM used in our prior experiments. To study the physiological and biotechnological relevance of EET in food fermentations, kale juice was fermented using *L. plantarum* as a starter culture (*Figure 6A*). The fermentation of kale juice was measured under EET conditions (a polarized anode with or without DHNA), and an OC control (a nonpolarized anode with DHNA) was used to separate the role of DHNA and electron flow to the anode on the fermentation process. An additional bioreactor without cells, but with DHNA, was operated to identify any possible electrochemical-driven conversion of substrates. When *L. plantarum* was added to the prepared kale juice, approximately 10-fold more current was generated during EET conditions with DHNA (EET+ DHNA) as compared to abiotic and biological non-EET promoting conditions (no DHNA) (*Figure 6B*). This current was comparable to the current generated in laboratory medium (*Figure 3C*), indicating that robust EET by LAB is possible in the complex physiological conditions of a food fermentation.

We next investigated the impact of EET on *L. plantarum* growth and metabolism in the kale juice fermentation. Significant changes in the pH and fermentation products were detected under EET conditions (*Figure 6C–D*). These differences occurred in the absence of significant changes in viable cell numbers (*Figure 6—figure supplement 1A*) at the time points measured. As previously observed using laboratory culture media, an approximately 2-fold greater accumulation of total end-fermentation products per cell was obtained when cells interacted with an anode in the presence of DHNA (*Figure 6C*). In the kale juice fermentation, EET+ DHNA conditions enhanced both lactate and acetate production per cell without changing the distribution of metabolites (*Figure 6E* and *Figure 6—figure supplement 1B*). Thus, when DHNA was provided, EET enhanced the overall yield of fermentation end-products and their production rates per cell, mimicking our observations in laboratory medium (*Figure 5C–D*). EET also led to a significantly higher acidification of the kale juice compared to OC conditions, and the presence of DHNA dramatically enhanced this pH drop (*Figure 6D*). In general, when no DHNA was supplied but an anode was present as an electron acceptor, the fermentation process was very similar to OC conditions. This means in kale juice, a source of quinones is essential to support *L. plantarum* EET activity. Overall, these results show that EET under physiological

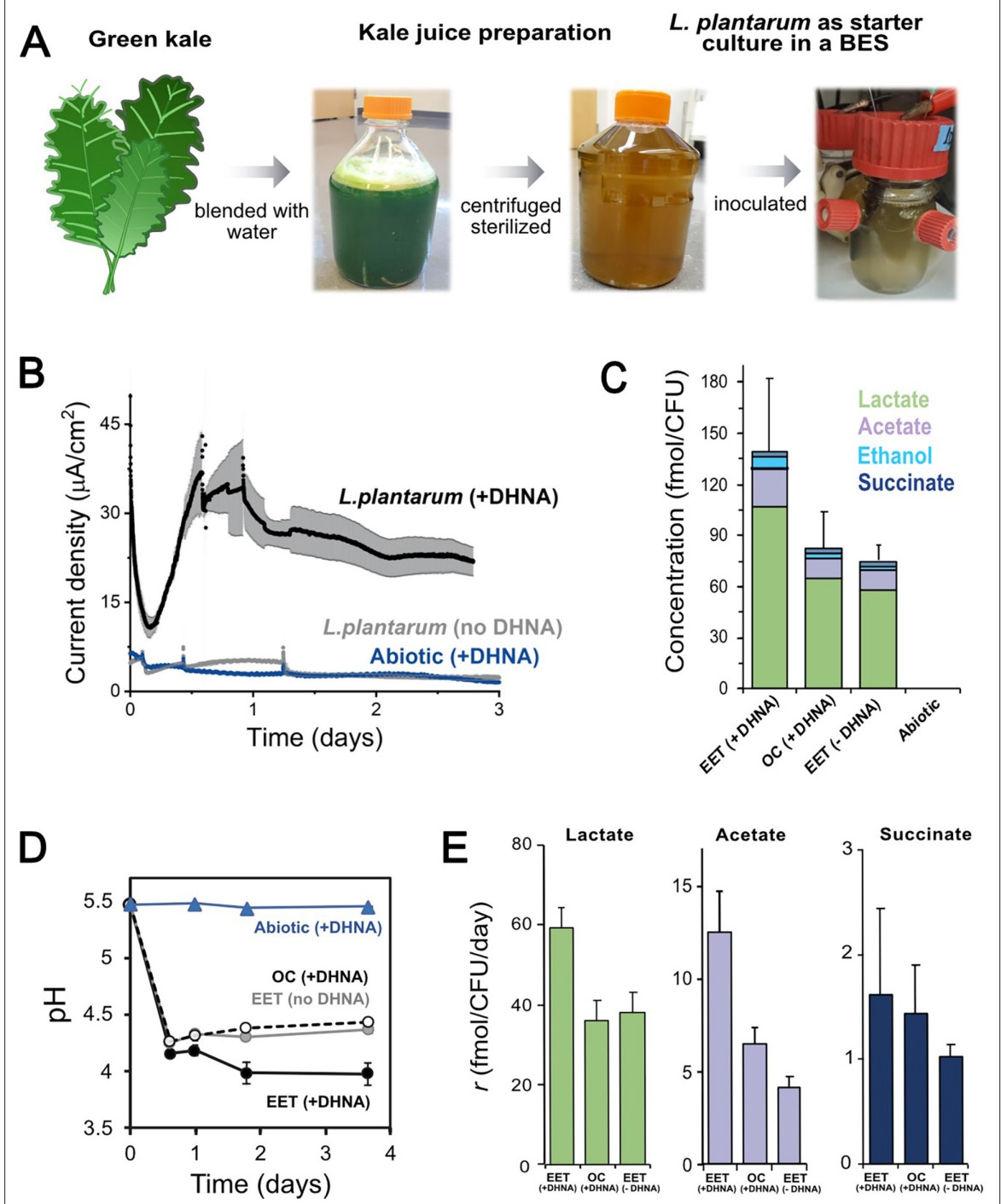

**Figure 6.** EET in a kale juice increases the production of fermentation end products. (**A**) Preparation of kale juice medium used for fermentation in bioelectrochemical reactors. (**B**) Current density production measured from kale juice medium over time in the presence of *L. plantarum* and 20 µg/mL DHNA, no DHNA, or under abiotic conditions with addition of 20 µg/mL DHNA. The anode polarization was maintained at 0.2 $V_{Ag/AgCl}$. (**C**) Normalized total quantities of the metabolites detected per cell ($CFU_{max}$ used for calculations). (**D**) pH measurements over time under different conditions tested on a second set of kale juice fermentations performed under the same conditions. (**E**) Production rate per viable cell, *r*, of lactate, acetate, and succinate. The avg ± stdev of three biological replicates is shown. See also *Figure 6—figure supplement 1* and related data in *Figure 6—source data 1*.

The online version of this article includes the following source data and figure supplement(s) for figure 6:

**Source data 1.** Raw data of *Figure 6B–E*.

**Figure supplement 1.** EET does not impact cell viability and distribution of metabolites in a kale fermentation.

**Figure supplement 1—source data 1.** Raw data of *Figure 6—figure supplement 1A, B*.

conditions impacts cellular metabolism in *L. plantarum* by increasing metabolic flux which ultimately can affect the flavor profile of fermented foods (*Chen et al., 2017*).

## Discussion

Increases in fermentation and energy conservation from EET have important bioenergetic implications for the mainly fermentative LAB. We showed that *L. plantarum* and other diverse LAB species perform EET if riboflavin and quinones are present. *L. plantarum* EET activity requires an NADH dehydrogenase (Ndh2) and conditionally requires an extracellular, flavin-binding reductase (PplA). EET in *L. plantarum* generates a high $NAD^+/NADH$ ratio, increases fermentation yield and flux, shortens lag phase, and increases ATP production. Thus, EET in *L. plantarum* is a hybrid energy metabolism that contains metabolic features of fermentation, redox features of anaerobic respiration, and predominately uses substrate-level phosphorylation to conserve energy. This pathway is active in *L. plantarum* with physiologically relevant DHNA concentrations and in a food fermentation and results in an increased metabolic flux and acidification rate.

### The combined EET fermentation hybrid metabolism is distinct from anaerobic respiration, fermentation, and other energy conservation strategies

When performing EET, the metabolism of *L. plantarum,* a primarily fermentative bacterial species, is fundamentally different from EET-driven, anaerobic respiration of metal-reducing bacteria. Although aspects of EET in *L. plantarum* and metal-reducing *Geobacter* spp. are similar, such as the upregulation of NADH dehydrogenase, the reduction rate of extracellular electron acceptors, and the high $NAD^+/NADH$ ratio, other aspects of energy metabolism during EET in these two organisms are starkly different (see comparison in *Supplementary file 2*). *Geobacter* spp. direct their metabolic flux through the TCA cycle, rely almost exclusively on extracellular electron acceptors to regenerate NADH, and produce ATP exclusively through oxidative phosphorylation. In contrast, *L. plantarum* regenerates a substantial fraction of its NADH by directing metabolic flux through fermentative pathways. Additionally, oxidative phosphorylation is not a major mechanism of energy conservation in *L. plantarum* during EET, as supported by three lines of evidence: the marginal metabolic flux through the reduced branch of TCA cycle, no involvement of known PMF-generating proteins, and that increased ATP levels can be accounted for by increased substrate-level phosphorylation. While additional data are required to eliminate the possibility that oxidative phosphorylation is occurring in *L. plantarum* during EET, we can qualitatively state that substrate-level phosphorylation is the major mechanism for ATP generation.

Comparing EET and respiratory metabolism in LAB also reveals substantial differences in these metabolisms (*Supplementary file 2*). While both metabolisms require quinones, respiration also requires exogenous heme. Our findings and similar findings in *E. faecalis* (*Pankratova et al., 2018*) confirm that heme is not required for EET. Moreover, EET also differs from respiration in LAB because it occurs at the start of or prior to exponential phase growth, does not change the final cell density, and increases fermentation with no effect on the resultant proportions of lactate, acetate, and ethanol (*Duwat et al., 2001*). Thus, EET in LAB diverges from respiration in metal-reducing bacteria or LAB in its metabolic pattern and energetic consequences. In addition, while EET provokes a shift in fermentative metabolism in other bacteria upon the addition of artificial mediators (*Vassilev et al., 2021*; *Emde and Schink, 1990*), *L. plantarum* EET is active upon the presence of a mediator present in a complex food system.

This EET mechanism is also a novel energy conservation strategy compared to known fermentative metabolisms in LAB (comparison in *Supplementary file 2*). *L. plantarum* and other LAB, reduce alternative intracellular electron acceptors like citrate, fructose, and phenolic acids, resulting in increased intracellular $NAD^+/NADH$ ratios (*Hansen, 2018*). This metabolic activity is especially important for heterofermentative LAB in order to synthesize additional ATP through acetate kinase (*Gänzle, 2015*). Unlike these examples, however, the reduction of extracellular $Fe^{3+}$ or an anode by EET requires a respiratory protein (Ndh2) and the shuttling of electrons outside of the cell. In addition, the reduction of the oxygen and organic compounds for cofactor regeneration by LAB leads to a metabolic shift toward acetate production and altered metabolic end-product ratios (*Gänzle, 2015*), which does not

occur during EET. These differences show how the hybrid metabolism under EET conditions is distinct from other pathways that alleviate reduced intracellular conditions in LAB.

Previous studies have reported a simultaneous use of fermentation and electron transport elements, such as in respiro-fermentation in *Saccharomyces cerevisiae* (*Blom et al., 2000*). However, respiro-fermentation produces ATP and maintains intracellular redox balance through substrate-level phosphorylation and/or oxidative phosphorylation using separate pathways. Our data strongly suggests a single pathway is responsible for both ATP generation and intracellular redox balance. This hybrid fermentation mode is also different from the electron bifurcating mechanism, in which the extra ATP generation is driven by the creation of a $H^+$ or $Na^+$ potential from the oxidation of a ferredoxin (*Buckel and Thauer, 2013*). Unlike in this example, EET in LAB does not involve PMF creating elements and EET drives ATP generation through substrate-level phosphorylation. Another poorly understood example of the use of substrate-level phosphorylation and electron transport chains to balance intracellular redox state is found in the non-fermentative bacterium *S. oneidensis*. Although this species is a respiratory bacterium, it relies predominately on substrate-level phosphorylation to grow anaerobically with the exogenous electron acceptor fumarate (*Hunt et al., 2010*). In this scenario, it is unclear if there are changes in intracellular redox state or metabolism in this species.

In contrast to and expanding upon these studies, our work elucidates a qualitatively and quantitatively different blending of fermentation and respiration. *L. plantarum* EET-associated metabolism contains features of both fermentation (e.g. substrate-level phosphorylation, high fermentation product yields) and respiratory metabolisms (e.g. $NAD^+/NADH$ ratios, NADH dehydrogenase required) (*Supplementary file 2*). Quantitatively, this hybrid metabolism leads to an overall ~1.75x-more efficient and ~1.75x-faster energy conservation (increased $Y_{mannitol}$, mannitol flux), but an overall ~1.5 fold weaker coupling between anabolism and catabolism (lower $Y_{ATP}$). Additionally, the increased $NAD^+/NADH$ ratio arises from using an ~2:1 ratio of endogenous to extracellular electron acceptors. Thus, to our knowledge, the hybrid strategy that *L. plantarum* uses to generate ATP performing EET constitutes a novel mode of energy conservation in a primarily fermentative microorganism.

## Different mechanisms of EET appear to be widespread in LAB

Based on our observations and others, we propose that EET is widespread in LAB and occurs by different mechanisms. Besides *L. plantarum*, we showed that *L. lactis* is able to generate current despite lacking *pplA*. Current generation by *L. lactis* was observed previously, found to be riboflavin dependent, and resulted in a small metabolic shift (yet to be defined) in which the flux through NADH-oxidizing pathways was reduced and ATP generating pathways were increased (*Freguia et al., 2009*; *Masuda et al., 2010*). *L. lactis* can also perform EET by reduction of tetrazolium violet and this activity depends on the presence of both quinones and an NADH dehydrogenase (NoxAB) (*Tachon et al., 2009*). *E. faecalis* is another LAB that performs EET, and similar to *L. plantarum*, it requires quinones (*Pankratova et al., 2018*) and a type-II NADH dehydrogenase (Ndh3) (*Hederstedt et al., 2020*) for $Fe^{3+}$ reduction. In contrast to this mechanistic similarity, *E. faecalis* performs EET using matrix-associated iron resulting in both increased final cell biomass and intracellular iron (*Keogh et al., 2018*). Moreover, unlike *L. plantarum* and *L. lactis*, the presence of PplA is not necessary for anode reduction or $Fe^{3+}$ reduction (*Hederstedt et al., 2020*). The conditional need for PplA in EET may be explained by the different prior growth conditions used and/or related to the existence of different mechanisms and proteins depending on the redox potential of the extracellular electron acceptor. Other flavin-binding, extracellular reductases amongst Gram-positive organisms, such as FrdA (acting on fumarate) have been identified in *L. monocytogenes* and UrdA (acting on urocanate) in *Enterococcus rivorum* (*Light et al., 2019*). Thus, there may exist a yet unidentified extracellular reductase in *L. plantarum* and *L. lactis* required for anode reduction. Thus, our findings elucidate a new pattern of metabolic changes associated with EET. It seems likely that these many mechanisms reflect the ability of EET to alleviate constraints of intracellular redox balance in fermentative metabolism across LAB.

## EET has important implications for ecology and biotechnology of LAB

Conservation of the FLEET locus among different LAB species supports the premise that this hybrid fermentation with EET provides an important metabolic strategy for these bacteria in their natural habitats. LAB with a complete FLEET locus are homofermentative, thus underscoring the distinct ways homofermentative and heterofermentative LAB have evolved for energy conservation (*Salvetti*

*et al., 2013*). *L. plantarum* and other LAB with FLEET systems such as *L. casei* are genetically and metabolically diverse and grow in a variety of nutrient rich environments including dairy and plant foods and the digestive tract (*Cai et al., 2009*; *Martino et al., 2016*; *Siezen et al., 2010a*). Those environments also are rich sources of sources of quinones, flavins, and extracellular electron acceptors such as iron (*Cataldi et al., 2003*; *Fenn et al., 2017*; *Kim, 2017*; *Roughead and McCormick, 1990*; *Walther et al., 2013*). Increased organic acid production and environmental acidification by LAB with this hybrid metabolism would provide an effective mechanism to inhibit competing microorganisms and confer a competitive advantage for growth. The increased ATP relative to biomass generation observed during growth on mannitol might also give sufficient readiness for using this energy later on to outcompete neighboring organisms (*Russell and Cook, 1995*). These effects of EET may be particularly important on plant tissues and intestinal environments, wherein LAB tend to be present in low numbers. Besides our observation that *L. plantarum* performs EET in kale juice, the FLEET pathway is important for intestinal colonization by both *L. monocytogenes* (*Light et al., 2018*) and *E. faecalis* (*Lam et al., 2019*), and *L. plantarum* FLEET genes including *ndh2* and *pplA* were highly induced in the small intestine of rhesus macaques (*Golomb et al., 2016*).

The hybrid fermentation metabolism of LAB also has technological relevance. For many LAB food fermentations, acidification of the food matrix is required to prevent the growth of undesired microorganisms and result in a more consistent and reproducible product (*Marco et al., 2021*). Starter cultures are frequently selected based on their capacity for rapid growth and acid production (*Bintsis, 2018*). In the presence of an anode, exposure of *L. plantarum* to EET conditions during kale juice fermentation increased the acidification rate. Thus, this shows that EET metabolism is active in complex nutritive environments such as kale leaf tissues that contain other potential electron acceptors besides the anode and diverse electron donors (glucose, fructose, sucrose) (*Thavarajah et al., 2016*). This example also shows how electro-fermentation, the technological process by which fermentation is modulated using electrodes, can be used to control food fermentations (*Moscoviz et al., 2016*; *Schievano et al., 2016*; *Vassilev et al., 2021*). Because *L. plantarum* also increased fermentation flux when the electrode was available as an electron sink, higher quantities of organic acid flavor compounds were formed. Therefore, by the manipulation of extracellular redox potential, food electro-fermentations may be used to control microbial growth. This would allow the creation of new or altered sensory profiles in fermented foods, such as through altered organic acid production and metabolism or synthesis of other compounds that alter food flavors, aromas, and textures.

## Final perspective

We expect that our study will improve the current understanding of energy conservation in primarily fermentative microorganisms and contribute to establishing the ecological relevance of EET in lactic acid bacteria. This work will ultimately allow the use of EET to electronically modulate the flavor and textural profiles of fermented foods and expand the use of lactic acid bacteria in bioelectronics, biomedicine, and bioenergy (*Moscoviz et al., 2016*). The identification of the precise components and full bioenergetics involved in *L. plantarum* EET will be key to unravel physiological and ecological questions and to develop other biotechnological applications.

# Materials and methods

### Key resources table

| Reagent type (species) or resource | Designation | Source or reference | Identifiers | Additional information |
|---|---|---|---|---|
| Strain, strain background (*Lactiplantibacillus plantarum*) | NCIMB8826 | *Dandekar, 2019* | | Strain information listed in *Supplementary file 3* |
| Strain, strain background (*Lactiplantibacillus plantarum*) | NCIMB8826-R | *Yin et al., 2018* | Rifampicin-resistant mutant of NCIMB8826 | Strain information listed in *Supplementary file 3* |
| Strain, strain background (*Lactiplantibacillus plantarum*) | MLES100 | This study | Deletion mutant of NCIMB8826 lacking *ndh2* | Plasmid information listed in *Supplementary file 3* |
| Strain, strain background (*Lactiplantibacillus plantarum*) | MLES101 | This study | Deletion mutant of NCIMB8826 lacking *pplA* | Plasmid information listed in *Supplementary file 3* |

*Continued on next page*

*Continued*

| Reagent type (species) or resource | Designation | Source or reference | Identifiers | Additional information |
|---|---|---|---|---|
| Strain, strain background (*Lactiplantibacillus plantarum*) | MLEY100 | This study | Deletion mutant of NCIMB8826 lacking *narGHIJ* | Plasmid information listed in *Supplementary file 3* |
| Strain, strain background (*Lactiplantibacillus plantarum*) | B1.3 | *Yin et al., 2018* | | Strain information listed in *Supplementary file 3* |
| Strain, strain background (*Lactiplantibacillus plantarum*) | AJJ11 | *Yu et al., 2021* | | Strain information listed in *Supplementary file 3* |
| Strain, strain background (*Lactiplantibacillus plantarum*) | 8.1 | *Yu et al., 2021* | | Strain information listed in *Supplementary file 3* |
| Strain, strain background (*Lactiplantibacillus plantarum*) | ATCC 202195 | *Wright et al., 2020* | | Strain information listed in *Supplementary file 3* |
| Strain, strain background (*Lactiplantibacillus plantarum*) | NCIMB700965 | *Heeney and Marco, 2019* | | Strain information listed in *Supplementary file 3* |
| Strain, strain background (*Lactiplantibacillus pentosus*) | BGM48 | *Golomb et al., 2013* | | Strain information listed in *Supplementary file 3* |
| Strain, strain background (*Lactiplantibacillus casei*) | BL23 | *Mazé et al., 2010* | | Strain information listed in *Supplementary file 3* |
| Strain, strain background (*Levilactobacillus brevis*) | ATCC 367 | *Makarova et al., 2006* | | Strain information listed in *Supplementary file 3* |
| Strain, strain background (*Lactococcus lactis*) | KF147 | *Siezen et al., 2010b* | | Strain information listed in *Supplementary file 3* |
| Strain, strain background (*Lactococcus lactis*) | IL1403 | *Bolotin et al., 2001* | | Strain information listed in *Supplementary file 3* |
| Strain, strain background (*Lactiplantibacillus Rhamnosus*) | GG | *Kankainen et al., 2009* | | Strain information listed in *Supplementary file 3* |
| Strain, strain background (*Lactiplantibacillus murinus*) | ASF361 | *Wannemuehler et al., 2014* | | Strain information listed in *Supplementary file 3* |
| Strain, strain background (*Enterococcus faecalis*) | ATCC 29212 | *Minogue et al., 2014* | | Strain information listed in *Supplementary file 3* |
| Strain, strain background (*Enterococcus faecalis*) | ATCC 8459 | *Kopit et al., 2014* | | Strain information listed in *Supplementary file 3* |
| Strain, strain background (*Pediococcus pentosaceus*) | ATCC 25745 | *Makarova et al., 2006* | | Strain information listed in *Supplementary file 3* |
| Strain, strain background (*Streptococcus agalactiae*) | ATCC 27956 | *McDonald and McDonald, 1976* | | Strain information listed in *Supplementary file 3* |
| Strain, strain background (*Escherichia coli*) | DH5α | *Taylor et al., 1993* | *fhuA2 lac(del)U169 phoA glnV44 Φ80′ lacZ(del) M15 gyrA96 recA1 relA1 endA1 thi−one hsdR17*, amplification of cloning vector | |

## Strains and culture conditions

All strains and plasmids used in this study are listed in *Supplementary file 3*. Standard laboratory culture medium was used for routine growth of bacteria as follows: *Lactiplantibacillus* spp., *Lacticaseibacillus* spp., *Levilactobacillus brevis*, *Ligilactobacillus murinus*, and *Pediococcus pentosaceus*, MRS (BD, Franklin Lakes, NJ, USA); *Lactococcus lactis* and *Streptococcus agalactiae*, M17 (BD) with 2% w/v glucose; *Enterococcus faecalis*, and *Enterococcus faecium*, BHI (BD); and *Escherichia coli*, LB (Teknova, Hollister, CA, USA). Bacterial strains were incubated without shaking except for *E. coli* (250 RPM) and at either 30 or 37 °C. Where indicated, strains were grown in filter-sterilized MRS (*De MAN et al., 1960*) lacking beef extract with either 110 mM glucose [gMRS] or 110 mM mannitol [mMRS], or a chemically defined minimal medium (*Supplementary file 4*) with 125 mM glucose [gCDM] or 125 mM mannitol [mCDM] for 18 hr (*Aumiller et al., 2021*). Riboflavin (1 mg/L) was routinely added to the CDM. When indicated, culture medium was supplemented with 20 µg/mL of the quinone 1,4-dihydroxy-2-naphthoic acid (DHNA) (Alfa Aesar, Haverhill, MA, USA), 1.25 mM ferric ammonium citrate ($C_6H_8FeNO_7$) (1.25 mM) (VWR, Radnor, PA, USA), riboflavin (Sigma-Aldrich, St. Louis, MO, USA), or 5 µg/mL erythromycin (VWR).

## DNA sequence analysis

The FLEET gene locus of *L. plantarum* NCIMB8826 was identified using NCBI basic local alignment search tool (BLAST) (*McGinnis and Madden, 2004*) using the *L. monocytogenes* 10403S FLEET genes (lmo2634 to lmo2641) as a reference. *L. plantarum* genes were annotated based on predicted functions within the FLEET pathwa (*Light et al., 2018*). FLEET locus genes were identified in other LAB by examining 1,788 complete Lactobacillales genomes available at NCBI (downloaded 02/25/2021). A local BLAST (ver 2.10.1) database containing these genomes was queried using tBLASTx with NCIMB8826 FLEET genes a reference. A gene was considered to be present in the Lactobacillales strain genome if the Bit-score was $\geq$50 and the E-value was $\leq 10^{-3}$ (*Pearson, 2013*). Heatmaps showing the percentage of strains in Lactobacillales genera and the *Lactobacillus*-genus complex (*Zheng et al., 2020*) identified to contain individual FLEET genes were visualized using the R-studio package ggplot2 (*Wickham, 2011*) with clustering done through UPGMA. The FLEET loci of *L. plantarum* strain 8.1 and NCIMB700965 were aligned to the NCIMB8826 genome in MegAlign Pro (DNAstar Inc, Madison, WI, USA).

## Insoluble iron reduction assays

Cells were collected by centrifugation at 10,000 g for 3 min, washed twice in phosphate-buffered saline (PBS), pH 7.2 (http://cshprotocols.cshlp.org), and adjusted to an optical density (OD) at 600 nm ($OD_{600nm}$) of 2 in the presence of 2.2 mM ferrihydrite (*Schwertmann and Fischer, 1973*; *Stookey, 2002*) and 2 mM ferrozine (Sigma-Aldrich). Where indicated, 55 mM glucose or mannitol, 20 µg/mL DHNA, and riboflavin were added. After 3 hr incubation at 30 °C, the cells were collected by centrifugation at 10,000 g for 5 min and the absorbance of the supernatant was measured at 562 nm with a Synergy 2 spectrophotometer (BioTek, Winooski, VT, USA). Quantities of ferrihydrite reduced were determined using a standard curve containing a 2-fold range of $FeSO_4$ (Sigma-Aldrich) (0.25 mM to 0.016 mM) and 2 mM ferrozine. The $FeSO_4$ was dissolved in 10 mM cysteine-HCl (RPI, Mount Prospect, IL, USA) to prevent environmental re-oxidation of $Fe^{2+}$ to $Fe^{3+}$ in the standard curve. For testing iron reduction activity of cells with a DHNA concentration of 0.01 µg/mL in the medium, iron(III) oxide nanoparticles < 50 nm (Sigma-Aldrich) were used as insoluble iron form (*Figure 1—figure supplement 3*).

## *L. plantarum* mutant construction

*L. plantarum* NCIMB8826 *ndh2*, *pplA*, and *narGHIJ* deletion mutants were constructed by double-crossover homologous recombination with the suicide plasmid pRV300 (*Leloup et al., 1997*). For mutant construction, upstream and downstream flanking regions of these genes were amplified using the A/B and C/D primers, respectively, listed in *Supplementary file 5*. Splicing-by-overlap extension (SOEing) PCR was used to combine PCR products as previously described (*Heckman and Pease, 2007*). PCR products were digested with restriction enzymes EcoRI, SacI, SacII, or SalI (New England Biolabs, Ipswich, MA, USA) for plasmid ligation and transformation into *E. coli* DH5α. The resulting plasmids were then introduced to *L. plantarum* NCIMB8826 by electroporation. Erythromycin-resistant mutants were selected and confirmed for plasmid integration by PCR (see *Supplementary file 5* for primer sequences). Subsequently, deletion mutants were identified by a loss of resistance to erythromycin, PCR (see *Supplementary file 5* for primer sequences) confirmation, and DNA sequencing (http://dnaseq.ucdavis.edu).

## Bioelectrochemical reactors (BES) construction, operation, and electrochemical techniques

*L. plantarum* NCIMB8826 strains were grown overnight (approximately 16–18 hr) from glycerol stocks in MRS. Cells were harvested by centrifugation (5200 g, 12 min, 4 °C) and washed twice in PBS. When *L. plantarum* wild-type EET activity versus the Δ*ndh2* (MLES100) and Δ*pplA* (MLES101) deletion mutants was compared, cells were grown as described and the number of cells was normalized across the three strains prior to inoculation in the BES. The bioreactors consisted of double-chamber electrochemical cells (Adams & Chittenden, Berkeley, CA) (*Figure 1B*) with a cation exchange membrane (CMI-7000, Membranes International, Ringwood, NJ) that separated them. A three-electrode configuration was used consisting of an Ag/AgCl sat KCl reference electrode (BASI, IN, USA), a titanium wire counter electrode, and a 6.35-mm-thick graphite felt working electrode (anode) of 4 × 4 cm (Alfa Aesar, MA, USA) with a piece of Ti wire threaded from bottom to top as a current collector and connection to

the potentiostat. We used a Bio-Logic Science Instruments (TN, USA) potentiostat model VSP-300 for performing the electrochemical measurements (chronoamperometry). The bioreactors were sterilized by filling them with ddH$_2$O and autoclaving at 121 °C for 30 min. The water was then removed and replaced with 150 mL of filter sterilized mMRS or mCDM media for the working electrode chamber, and 150 mL of M9 medium (6.78 g/L Na$_2$HPO$_4$, 3 g/L KH$_2$PO$_4$, 0.5 g/L NaCl, 1 g/L NH$_4$Cl) (BD) for the counter electrode chamber. Both media of the working electrode chamber were supplemented with 20 µg/mL DHNA or 0.01 µg/mL diluted 1:1 in DMSO:ddH$_2$O where appropriate. To test the role of *bd*-cytochrome, heme was added in a final concentration of 10 µg/mL (diluted 1:1 in DMSO: ddH$_2$O). The medium in the working electrode chamber was continuously mixed with a magnetic stir bar and N$_2$ gas was purged to maintain anaerobic conditions for the course of the experiment. The applied potential to the working electrode was of +0.2 V versus Ag/AgCl (sat. KCl) (BASI, IN, USA). Reactors run under OC conditions were similarly assembled but kept at open circuit and used as control for non-current circulating conditions. Once the current stabilized, the electrochemical cells were inoculated to a final OD$_{600}$ of 0.12–0.15 with the cell suspensions prepared in PBS. Current densities are reported as a function of the geometric surface area of the electrode (16 cm$^2$). The bioreactors were sampled by taking samples under sterilized conditions at different time points for subsequent analysis. The samples for organic acids analyses were centrifuged (15,228 g, 7 min) and the supernatant was separated for High-Performance Liquid Chromatography (HPLC) assessments. Samples for ATP and NAD$^+$/NADH analyses were flash frozen in a dry ice/ethanol bath.

## Metabolite analysis

Organic acids, ethanol, and sugar concentrations were measured by HPLC (Agilent, 1260 Infinity), using a standard analytical system (Shimadzu, Kyoto, Japan) equipped with an Aminex Organic Acid Analysis column (Bio-Rad, HPX-87H 300 × 7.8 mm) heated at 60 °C. The eluent was 5 mM of sulfuric acid, used at a flow rate of 0.6 mL min$^{-1}$. We used a refractive index detector 1260 Infinity II RID. A five-point calibration curve based on peak area was generated and used to calculate concentrations in the unknown samples. The following standards were included in the HPLC measurements: acetate, formate, pyruvate, malate, lactate, succinate, oxalacetate, fumarate, ethanol, acetoin, butanediol, mannitol, and glucose. No gaseous products were measured.

## BES biomass growth determination

Bioreactors were shaken to remove the cells attached to the working electrode and afterwards sampled to measure viable cells (colony forming units [CFUs]) and total biomass (dry weight). Samples for CFU enumeration were collected under sterile conditions at the time of inoculation and at the time of approximately maximum current density. Samples were serially diluted (1:1000 to 1:1000000) in sterile PBS and plated on MRS for CFUs enumeration after overnight incubation at 30 °C. Dry weight was determined using a 25 mL sample collected at approximately maximum current density. Cells were harvested by centrifugation (5250 g, 12 min, 4 °C) and washed twice in 50 mL ddH$_2$O. Afterwards cells were resuspended in 1 mL of ddH$_2$O and transferred to microfuge tubes (previously weighted). Cells were harvested by centrifugation (5250 g, 12 min, 4 °C), and the tubes were then transferred to an evaporator to remove humidity. The microfuge tubes were then cooled in a desiccator for 30 min and the weight of each tube was measured to determine cell weight. The difference between the weight of each tube with the pellet and before containing it allowed us to determine the dry weight/mL.

## RNA-seq library construction and transcriptome analysis

*L. plantarum* NCIMB8826 was grown in triplicate to exponential phase (OD$_{600}$ 1.0) at 37 °C in mMRS with or without the supplementation of 20 µg/mL DHNA and 1.25 mM ferric ammonium citrate. Cells were collected by centrifugation at 10,000 g for 3 min at 4 °C, flash frozen in liquid N$_2$ and stored at –80 °C prior to RNA extraction as previously described (*Golomb et al., 2016*). Briefly, frozen cell pellets were resuspended in cold acidic phenol:chloroform:isoamyl alcohol (pH 4.5) [125:24:1] (Invitrogen, Carlsbad, CA, USA) before transferring to 2 mL screw cap tubes containing buffer (200 mM NaCl, 20 mM EDTA), 20% SDS, and 300 mg 0.1 mm zirconia/silica beads. RNA was extracted by mechanical lysis with an MP Fastprep bead beater (MP Biomedicals, Santa Ana, CA, USA) at 6.5 m/s for 1 min. The tubes were centrifuged at 20,000 g at 4 °C for 3 min and the upper aqueous phase was

transferred to a new tube. The aqueous phase was extracted twice with chloroform:isoamyl alcohol [24:1] (Fisher Scientific, Waltham, MA, USA), The aqueous phase was then transferred to a new tube for RNA ethanol precipitation (*Green and Sambrook, 2020*). RNA was then quantified on a Nano-drop 2000c (ThermoFisher), followed by double DNAse digestion with the Turbo DNA-free Kit (Invitrogen) according to the manufacturer's protocols. The quality of the remaining RNA was checked using a Bioanalyzer RNA 6000 Nano Kit (Agilent Technologies, Santa Clara, CA, USA) (all RIN values > 9) and then quantified with the Qubit 2.0 RNA HS Assay (Life Technologies, Carlsbad, CA, USA). For reverse-transcription PCR (RT-PCR), 800 ng RNA was converted to cDNA with the High Capacity cDNA Reverse Transcription Kit (Applied Biosystems, Foster City, CA, USA) according to the manufacturer's protocols. Quantitative RT-PCR was performed on a 7,500 Fast Real-Time PCR System (Applied Biosystems) using the PowerUp SYBR Green Master Mix (ThermoFisher) and RT-PCR primers listed in *Supplementary file 5*. The 2-ΔΔCt method was used for relative transcript quantification using *rpoB* as a control (*Livak and Schmittgen, 2001*).

For sequencing, ribosomal-RNA (rRNA) was depleted from 4 µg RNA using the RiboMinus Eukaryote Kit v2 with specific probes for prokaryotic rRNA (ThermoFisher) following the manufacturer's instructions. RNA was then fragmented to approximately 200 bp, converted to cDNA, and barcoded using the NEBnext Ultra-directional RNA Library Kit for Illumina (New England Biolabs, Ipswitch, MA, USA) with NEBnext Multiplex Oligos for Illumina (Primer Set 1) (New England Biolabs) following the manufacturer's protocols. cDNA libraries containing pooled barcoded samples was run across two lanes of a HiSeq400 (Illumina, San Diego, CA, USA) on two separate runs for 150 bp paired-end reads (http://dnatech.genomecenter.ucdavis.edu/). An average of 36,468,428 raw paired-end reads per sample was collected (*Supplementary file 6*). The DNA sequences were quality filtered for each of the 12 samples by first visualizing with FastQC (ver. 0.11.8) (*Andrews, 2010*) to check for appropriate trimming lengths, followed by quality filtering with Trimmomatic (ver. 0.39) (*Bolger et al., 2014*). Remaining reads then were aligned to the NCIMB8826 chromosome and plasmids using Bowtie2 (ver. 2.3.5) in the [-sensitive] mode (*Langmead and Salzberg, 2012*). The resulting '.sam' files containing aligned reads from Bowtie2 were converted to '.bam' files with Samtools (ver 1.9) (*Li et al., 2009*) before counting aligned reads with FeatureCounts in the [-- stranded = reverse] mode (ver. 1.6.4) (*Liao et al., 2014*). Reads aligning to noncoding sequences (e.g. rRNA, tRNA, trRNA, etc.) were excluded for subsequent analyses. Differential gene expression based on culture condition was determined with DESeq2 (*Love et al., 2014*) using the Wald test in the R-studio shiny app DEBrowser (ver 1.14.2) (*Kucukural et al., 2019*). Differential expression was considered significant with a False-discovery-rate (FDR)-adjusted $p$-value $\leq 0.05$ and a $Log_2$ (fold-change) $\geq 0.5$. Clusters of Orthologous Groups (COGs) were assigned to genes based on matches from the eggNOG (ver. 5.0) database (*Huerta-Cepas et al., 2019*).

## Redox probe assays

Hamilton oxidation-reduction potential (ORP) probes (Hamilton Company, Reno, NV, USA) were inserted into air-tight Pyrex (Corning Inc, Corning, NY, USA) bottles containing mMRS supplemented with 20 µg/mL DHNA and/or 1.25 mM ferric ammonium citrate and incubated in a water bath at 37 °C. A custom cap for the Pyrex bottles was 3D printed with polylactic acid filament (2.85 mm diameter) such that the ORP probe threads into the cap and an o-ring seal can be used to provide an air-tight seal between the probe and the cap. The ORP was allowed to equilibrate over 40 min before *L. plantarum* NCIMB8826, Δ*ndh2* (MLES100), or Δ*pplA* (MLES101) were inoculated at an $OD_{600}$ of 0.10. Two uninoculated controls were used to measure baseline ORP over time. The ORP data was collected via Modbus TCP/IP protocol (Stride Modbus Gateway, AutomationDirect, Cumming, GA, USA) into a database (OSIsoft, San Leandro, CA, USA) and analyzed in MATLAB (Mathworks, Nantick, MA, USA). pH was measured using a Mettler Toledo SevenEasy pH meter (Mettler Toledo, Columbus, OH, USA). Cells were collected at either 24 hr or at the greatest ORP difference between the wild-type and mutant strains ($\Delta mV_{max}$) by centrifugation at 10,000 g for 3 min and used for ferrihydrite reduction analyses.

## ATP and NAD$^+$/NADH quantification

Frozen cell pellets were suspended in PBS and lysed by mechanical agitation in a FastPrep 24 (MP Biomedicals) at 6.5 m/s for 1 min. The cell lysates were then centrifuged at 20,000 g for 3 min at

4 °C. ATP and NAD$^+$ and NADH in the supernatants were then quantified with the Molecular Probes ATP Quantification Kit (ThermoFisher) and the Promega NAD/NADH-Glo Kit (Promega, Madison, WI, USA), respectively according to the manufacturers' instructions.

## Inductively coupled plasma-mass spectrometry (ICP-MS)

*L. plantarum* was inoculated in mMRS with or without 20 µg/mL DHNA and 1.25 mM ferric ammonium citrate at an $OD_{600}$ of 0.10 for 3.5 hr. Cells were then collected by centrifugation at 10,000 x g for 3 min and washed twice in PBS to remove cell-surface-associated metals. Viable cell numbers were enumerated by plating serial dilutions on MRS laboratory culture medium and the resulting cell materials were digested by incubating at 95 °C for 45 min in a 60% concentrated trace metal grade $HNO_3$, allowed to cool, then diluted with MilliQ water to a final concentration of 6% $HNO_3$. The contents were quantified with internal standards with an Agilent 7,500Ce ICP-MS (Agilent Technologies, Palo Alto, CA) for simultaneous determination of select metals (Na, Mg, Al, K, Ca, Cu, Zn, Ba, Mn, Fe) at the UC Davis Interdisciplinary Center for Plasma Mass Spectrometry (http://icpms.ucdavis.edu/).

## Kale juice fermentation assay

Green organic kale purchased from a market (Whole Foods) was washed with tap water and air dried for 1 hr as previously recommended (*Kim, 2017*). A total of 385 g of the leaves and stems were shredded with an electric food processor in 1 L ddH20. The kale juice was then diluted with 0.35 L ddH2O and autoclaved (121 °C, 15 min). The juice was then centrifuged under sterile conditions at 8000 rpm for 20 min and the supernatant was collected. A rifampicin-resistant variant of *L. plantarum* NCIMB8826-R (*Tachon et al., 2014*) (grown for 19 hr in MRS medium at 37 °C, 50 µg Rif/mL) was inoculated to an estimated final OD of approximately 0.05, and DHNA (20 µg/mL) was added where appropriate. Cells were collected and washed as previously described for the bioelectrochemical assays in mCDM. The anodic chambers of bioreactors assembled as previously described (anode of 4.3*6 cm) were filled with 125 mL of the inoculated kale juice and incubated at 30 °C purged with $N_2$. After 1 hr, the anodes were polarized to 0.2 V versus Ag/AgCl (sat. KCl) (EET conditions) or kept at open circuit (OC, no EET). Viable cells were measured by plating 10-fold serial dilutions in MRS agar plates with 50 µg/mL of Rif.

## Calculations

The total electrons harvested on the anode were estimated by integrating the area (charge) under the chronoamperometric curve (current response (A) over time (s)), which was corrected by subtracting the current baseline obtained before *L. plantarum* was added to the system. This obtained charge was then converted to mol of electrons using the Faraday constant (96,485.3 A*s/mol electrons).

## Data accession numbers

*L. plantarum* RNA-seq data are available in the NCBI Sequence Read Archive (SRA) under BioProject accession no. PRJNA717240. A list of the completed Lactobacillales genomes used in the DNA sequence analysis is available in the Harvard Dataverse repository at https://doi.org/107910/DVN/IHKI0C.

## Acknowledgements

This work was supported by the National Science Foundation grant #1650042, Office of Naval Research grant 0001418IP00037 (CMAF), and the USDA National Institute of Agriculture Multi-State Project (W4122). Work at the Molecular Foundry was supported by the Office of Science, Office of Basic Energy Sciences, of the U.S. Department of Energy under Contract No. DE-AC02-05CH11231. James Nelson was supported by the Rodgers University fellowship in Electrical and Computer Engineering.

## Additional information

### Funding

| Funder | Grant reference number | Author |
|---|---|---|
| National Science Foundation | 1650042 | Eric T Stevens |
| Office of Naval Research | 0001418IP00037 | Caroline M Ajo-Franklin |
| U.S. Department of Energy | DE-AC02-05CH11231 | Sara Tejedor-Sanz Eric T Stevens Caroline M Ajo-Franklin Maria L Marco |
| U.S. Department of Agriculture | W4122 | Maria L Marco |
| Cancer Prevention and Research Institute of Texas | RR190063 | Caroline M Ajo-Franklin Siliang Li |

The funders had no role in study design, data collection and interpretation, or the decision to submit the work for publication.

### Author contributions

Sara Tejedor-Sanz, Eric T Stevens, Conceptualization, Data curation, Formal analysis, Investigation, Methodology, Validation, Visualization, Writing – original draft, Writing – review and editing; Siliang Li, Formal analysis, Investigation, Validation, Visualization, Writing – review and editing; Peter Finnegan, Data curation, Investigation, Software, Writing – review and editing; James Nelson, Data curation, Investigation, Writing – review and editing; Andre Knoesen, Data curation, Methodology, Resources, Software, Supervision, Writing – review and editing; Samuel H Light, Conceptualization, Writing – review and editing; Caroline M Ajo-Franklin, Maria L Marco, Conceptualization, Funding acquisition, Methodology, Supervision, Writing – original draft, Writing – review and editing

### Author ORCIDs

Sara Tejedor-Sanz ⓘ http://orcid.org/0000-0002-1192-8256
Eric T Stevens ⓘ http://orcid.org/0000-0003-0648-8281
Samuel H Light ⓘ http://orcid.org/0000-0002-8074-1348
Caroline M Ajo-Franklin ⓘ http://orcid.org/0000-0001-8909-6712
Maria L Marco ⓘ http://orcid.org/0000-0002-3643-9766

### Decision letter and Author response

Decision letter https://doi.org/10.7554/eLife.70684.sa1
Author response https://doi.org/10.7554/eLife.70684.sa2

## Additional files

### Supplementary files

• Supplementary file 1. Data used for calculating the bioenergetic balances.

• Supplementary file 2. Comparison of the energy metabolism discovered in this study with fermentation in LAB and anaerobic respiration in *Geobacter* spp.

• Supplementary file 3. Strains and plasmids used in this study.

• Supplementary file 4. Chemically defined medium.

• Supplementary file 5. Primers developed for this study.

• Supplementary file 6. Transcriptome read counts, alignment rate, and gene assignment rate.

• Transparent reporting form

### Data availability

*L. plantarum* RNA-seq data are available in the NCBI Sequence Read Archive (SRA) under BioProject accession no. PRJNA717240. A list of the completed Lactobacillales genomes used in the DNA

sequence analysis is available in the Harvard Dataverse repository at https://doi.org/10.7910/DVN/IHKI0C All other data generated or analysed during this study are included in the manuscript and supporting files.

The following dataset was generated:

| Author(s) | Year | Dataset title | Dataset URL | Database and Identifier |
|---|---|---|---|---|
| Stevens E | 2021 | Lactiplantibacillus plantarum transcriptome under extracellular electron transfer (EET)-conducive conditions | https://www.ncbi.nlm.nih.gov/bioproject/PRJNA717240 | NCBI BioProject, PRJNA717240 |

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
