## [Editor Report]

In this study, the authors describe unique metabolic strategies, including extracellular electron transfer, utilized by the lactic acid bacterium *Lactiplantibacillus plantarum*. The ability to shift and/or accelerate metabolism of lactic acid bacteria capable of extracellular electron transfer may have interesting biotechnological applications.

---

## [Decision Letter]

**Decision letter after peer review:**

Thank you for submitting your article "Extracellular electron transfer increases fermentation in lactic acid bacteria via a hybrid metabolism" for consideration by *eLife*. Your article has been reviewed by 3 peer reviewers, and the evaluation has been overseen by a Reviewing Editor and Gisela Storz as the Senior Editor. The reviewers have opted to remain anonymous.

Essential revisions:

1) The methods and results of metabolite analyses are not sufficiently described. Further explanation is required if conclusions are to be made about yield, fermentation, and physiology.

2) Some claims are overstated and need to be revised or, better yet, backed up with more experimental data.

*Reviewer #1 (Recommendations for the authors):*

1) The only weakness: The methods and results of metabolite analyses are not sufficiently described. This requires explanation if conclusions are to be made about yield, fermentation, and physiology.

The paper provides strong evidence that electron diversion to external acceptors is an electron sink. However, while quantitative conclusions are made about fermentation, only ~30% of the mannitol can be accounted for in the EET condition that is the focus of this paper. Put another way, the paper is about where electrons go, but the majority of the electrons are missing.

Clarify if this strain can excrete products such as sugar polymers, sugars, pyruvate, formate, glycerol, acetoin, or butanediol. State if standards for these were included in the product analysis (an RI detector can pick up nearly all of these). Explain why formate isn't present (was it not measured?), as this would allow determination of flux through PFL vs. a pyruvate excretion, a carbon sink of acetoin, or butanediol. Could missing carbon be due to excretion of a partially oxidized sugar like fructose (which would also show up on an RI detector), or accumulation of storage polysaccharide?

A) Address these questions in the section around line 299; where fermentation products are mentioned as being present or absent. What was looked for, but not found (lactate yes, succinate no), vs which were not looked for, (pyruvate? Formate? Acetoin, etc). This makes it clear that carbon is not missing, just that the methods were unable to detect it.

B) Table 1 and all other tables making calculations need significant qualifiers-- since NADH flux is guessed from mannitol disappearance (the assumption of 3 NADH, which would not be true if sugars, glycerol, or pyruvate were made), and NADH fates + ATP yields are based on a small subset of fermentation products detected. Statements like line 347 "EET allowed cells to produce more fermentation products per mol mannitol" can't be made. Nor can one conclusively say what % of electrons went to electrodes, nor what the YATP is. If anything, the fact that the mix of lactate, acetate and ethanol do not shift dramatically to acetate production (which is typically the result of any system with an NADH sink, such as interspecies hydrogen transfer), is surprising and indicates something unexplained is going on.

Again, the lack of a carbon balance takes nothing away from qualitative conclusion that mannitol electrons are diverted to EET, or the strength of the EET evidence. The electrons at the electrode or in the iron are real and countable, this just limits the quantitative statements that can be made.

2) L136; Reported Geobacter current densities easily reach 1000 µA/cm2 for flat surfaces, not 100-200.

3) Figure 3B: Many questions about this experiment. It is the only time it appears, yet is used as a quantitative way to compare mutants.

a) State if the iron(III) in the medium was reduced by mutants during this assay-for redox potential to drop equally suggests that it was (which contradicts 3A).

b) Provide the final pH during these experiments-- pH can affect E{degree sign}'.

c) Lack of DHNA is always used as a reliable negative control in the other experiments, provide any data that shows if the addition of DHNA is essential for the mV drop in this assay, so we know it is also an assay of EET.

Lacking better evidence/controls, this is one of the weaker bits of data in the main body, especially next to the more robust Fe/current data. Appropriate for supplements, but not conclusive as is.

4) An idea regarding the ∆pplA phenotype-the two conditions require EET, but iron oxides should be lower potential acceptors than the anode which is poised at a high driving force of >+400 mV. The anode is plenty high to oxidize something like DHNA (E{degree sign}' ~70 mV) or chelated metals. PplA could be more necessary near the low potential of flavins. The supplied data can't be used to compare the effect of specific riboflavin concentrations on iron reduction rates (only extents) vs current production rates, but it might be available.

5) Line 311, More data would be needed to use succinate production as evidence for or against respiration. A mol of pyruvate could be excreted as succinate due to TCA cycle oxidation (indicating respiration), manifesting as excess electrons relative to the amount of succinate produced (as additional pyruvate would have to be used to make OAA). Conversely, succinate could be due to electron disposal via a reductive pathway (indicating fermentation). Lacking good carbon or electron recovery, neither can be ruled out, though the non-respiratory reductive pathway is most logical.

6) Table 1/line 403-: The low YATP in cells using EET is a possible 'different energy conversion strategy'. (Side note- an energy spilling/Russell mechanism would be an 'ATP-dissipating', not -accumulating). I'd suggest uncoupling of acid production from growth by low pH seems likely. In most strains, simple glycolytic pathway functions at low external pH, but complex protein synthesis and replication processes become inhibited, ATP from substrate turnover is needed to maintain pmf and internal pH, and high internal organic acids become toxic. Citable yield/pH data can be hard to locate as it comes from the 80's in the age of calling every lactic acid bacterium a Streptococcus, but since the non-EET cultures at neutral pH got close to the Bauchop and Elsden YATP, and EET cultures at pH 4 were a fraction of this, these data look like common behaviour of many lactic acid bacteria. In future comparisons, pH control could help estimate energy yields from EET. The first paper I found in a quick search to illustrate pH effects was: Russell and Dombrowski, DOI: 10.1128/aem.39.3.604-610.1980.

7) The value for Geobacter ATP/substrate yield in Table S2 seemed very high. Mahadevan says Geobacter produces 0.5 mol ATP/mol acetate when using extracellular acceptors (3x higher when using internal acceptors like fumarate). If I understand the adjustment for 6-carbon compounds, would this make the value for Geobacter using EET to be 1.5 mol ATP/mol 6-C substrate?

*Reviewer #2 (Recommendations for the authors):*

Line 21 – acronyms like this make our field more challenging to understand and are unnecessary. Does the work presented actually support 'energy conservation'? Growth seems to occur after EET activity takes place rather than with it as discussed below.

Line 27 – 'fermentation yield' referring to biomass yield or fermentation product yield?

Line 29, Line 45, Line 75, Line 86, Line 479 – blending fermentation and respiration has been known for over a decade in Shewanella (see Hunt et al., 2010, JBact). The description on Line 517 regarding the major finding from this reference is incorrect. PMF is not used by S. oneidensis anaerobically to generate ATP as demonstrated by the robust anaerobic growth of an ATP synthase mutant.

Line 61 – implied here is that LAB are auxotrophic for heme and menaquinone, but should be explicitly stated.

Page 8 (top) – the comparison to other organisms is not particularly helpful unless current per unit cell or unit biomass is quantified, nor does it seem particularly relevant to the overall theme of this manuscript.

Line 152 – predicted to encode

Line 182 – predicted to encode

Line 183 – under the condition tested.

Line 200 – isn't this already known from the *Listeria* work?

Line 206 – unclear what 'extracellular redox potential' is, unclear to me if any differences are significant in Figure 3B. I guess this is a measurement of Fe(II) / Fe(III) in the medium?

Figure 3D – did the authors include a strain that lacked the cluster?

Line 252 – unclear to me how ICP-MS can distinguish between iron inside a cell and associated with the cell surface.

Line 256 – have other metals been tested? Frequently manganese substitutes for iron in enzymes from non-iron dependent bacteria. Manganese oxide is also a known substrate for EET in Geobacter and Shewanella and could be nutritionally significant to LAB.

Line 262 – does this mean a large amount of the cells that grew are now dead?

Figure 4 (and Figure 1) – I don't understand why maximum current output is completely decoupled from when biomass spikes. At 24h (maximum current), basically no growth has occurred. At 2 days and 5 days, when biomass is the greatest (in the WT culture), very little current is produced. This seems like direct evidence that EET is not coupled with metabolism in these bacteria. The ATP measurements were made at 24h, where OD of the mutant is decreasing (thus less ATP would be expected). Same issue with NAD/NADH measurement.

Figure 4C – Lines are undefined in the figure legend. Also, growth should be plotted on a semi-log scale, see: https://schaechter.asmblog.org/schaechter/2018/07/why-you-must-plot-your-growth-data-on-semi-log-graph-paper.html

Figure 5 – based on the growth curve shown in Figure 4, the OC culture lags behind and does not reach maximum OD until day 5. What do these measurements look like at day 5 (or 6)? It seems likely that the authors are drawing conclusions from cells in two different states (stationary phase vs exponential phase) rather than focusing on the finding that having an external electron acceptor appears to accelerate the metabolism of cells.

Line 465 – Description of the experiment setup in 2B does not include addition of flavins for the iron reduction assay, so it is unclear why the authors claim that both flavins and quinones are necessary for EET in diverse LAB species.

Line 473 – Iron and electrode reduction is dependent on addition of DHNA. Are these cells likely to ever encounter 20 ug/mL DHNA in the environment? Is the EET activity observed relevant to normal physiology for these organisms, or a lab artifact caused by providing a non-native electron carrier? The argument beginning on line 491 underscores the artificial nature to me.

Line 559 – none of the references here provide any quantitation quinone concentration in the digestive tract. (Relates to the comment above from line 473).

Line 637 – is ferrozine toxic to these cells?

Line 646 – Were these mutants complemented?

Line 653 – the suicide vector should be identified and referenced here.

General comments:

Unclear why Figure 1C shows twice as much current twice as fast as Figure 3C in what appears to be identical conditions?

*Reviewer #3 (Recommendations for the authors):*

Figure 1A – though the tubes are aligned with the plot below, it may be helpful for some readers to have labels on this image to clarify the conditions of each.

Figure 1C – for the top line, the "+DHNA+mannitol" label should be on one line

Figure 4A – labeling should be more consistent ("anodic" vs. "EET")

Figure 4A – "log" should be removed from the y-axis label

Figure 4C – points on this plot should be connected with straight lines

Figure 5D – axis label is cut off

Figure 5G – FLEET should not be mentioned in the model at all. Instead, Ndh2 should be labeled

Figure 6A – Label differences between the two kale juice bottles

Authors should also provide more information for the definitions of YATP and the values in Table 1

Line 137: Where the authors compare current density capacity for *L. plantarum* to that of *S. oneidensis*--are the quoted values for *S. oneidensis* using (an) endogenously produced shuttle(s), or are they for *S. oneidensis* + (an) added shuttle(s)? This should be clarified, because *L. plantarum* has no current density capacity without added DHNA.

Authors should offer a mechanism for the conditional pplA phenotype

Switching between the terms "electron donor" and "carbon source" may be confusing for some readers; one term should be used consistently.

Figure 3-—figure supplement 1F is not discussed in the text.

Line 204: typo in "pplA"

The ndh2 mutant should be included in the kale juice fermentation experiments to test the contribution of Ndh2 in this context.

Readers may not be familiar with the term "Crabtree-positive"

"SLP" should be defined on page 3

The introduction does not provide enough detail for a broad audience to understand the metabolic pathways being discussed. It would help if some of the detail about *L. plantarum* metabolism (such as lines 318-321, 553-555) was moved from the results/discussion to the introduction. If *L. plantarum* is able to grow via respiration (as suggested in lines 304-308), this should also be mentioned in the introduction. The introduction should also be revised to ensure that terms are used unambiguously and to remove inaccurate generalizations. Some examples are given in the comments below.

It is not totally accurate to say that energy per se is generated or created during metabolism. The authors should revise these statements to say that energy is obtained or conserved or that ATP is generated.

This paper discusses detailed metabolic pathways and the terms used to describe these pathways should be used carefully and precisely to avoid confusion for the reader. For instance, the authors should precisely define "substrate-level phosphorylation" and "fermentation" and should not use these terms interchangeably. "Substrate-level phosphorylation" refers to a reaction that generates ATP directly, without the use of the proton motive force and the membrane-associated ATP synthase enzyme. "Fermentation" refers to the pathways that restore the NAD+/NADH ratio and allow growth with substrate-level phosphorylation as the sole mechanism of ATP generation.

Lines 50-51 and 373-374: Some organic compounds can also be respired (for example fumarate).

Lines 51-52 should say "Fermentative bacteria produce ATP solely via substrate-level phosphorylation" because substrate-level phosphorylation also occurs when bacteria are growing by respiration (i.e., in glycolysis).

Lines 71-73: It is unclear how the property of being "exergonic" or "endergonic" is relevant in the context here.

Lines 85-86: "…EET is associated with non-fermentative respiratory organisms…" Does this refer specifically to endogenous EET (i.e., without an added mediator?). This should be clarified because there is a long history of research on fermentation+EET in the presence of added mediators (see next comment).

Related to the above point: In the introduction or discussion, the authors should mention or acknowledge other work examining how growth in the presence of an electrode affects fermentative metabolisms. Some examples are:

Glasser NR, Kern SE, Newman DK. Phenazine redox cycling enhances anaerobic survival in *Pseudomonas aeruginosa* by facilitating generation of ATP and a proton-motive force. Mol Microbiol. 2014 Apr;92(2):399-412. doi: 10.1111/mmi.12566.

Emde R, Schink B. Enhanced Propionate Formation by Propionibacterium freudenreichii subsp. freudenreichii in a Three-Electrode Amperometric Culture System. Appl Environ Microbiol. 1990 Sep;56(9):2771-6. doi: 10.1128/aem.56.9.2771-2776.1990.

And other references discussed in reviews such as:

Moscoviz R, Toledo-Alarcón J, Trably E, Bernet N. Electro-Fermentation: How To Drive Fermentation Using Electrochemical Systems. Trends Biotechnol. 2016 Nov;34(11):856-865. doi: 10.1016/j.tibtech.2016.04.009.

Vassilev I, Averesch NJH, Ledezma P, Kokko M. Anodic electro-fermentation: Empowering anaerobic production processes via anodic respiration. Biotechnol Adv. 2021 May-Jun;48:107728. doi: 10.1016/j.biotechadv.2021.107728.

Comparing Figure 1A to Figure 3A, the extent of Fe3+ reduction seems quite different. Were the conditions the same for these two experiments? If so, an explanation should be added to the text.

Lines 104-106: Can DHNA itself be used as an electron shuttle? This should be addressed.

Line 137: Where the authors compare current density capacity for *L. plantarum* to that of *S. oneidensis*--are the quoted values for *S. oneidensis* using (an) endogenously produced shuttle(s), or are they for *S. oneidensis* + (an) added shuttle(s)? This should be clarified, because *L. plantarum* has no current density capacity without added DHNA.

Line 139: "current production was not dependent on the carbon source or growth medium, and current increased after supplementation of riboflavin in the growth medium". Both of these claims are hard to evaluate because the plots in figure 1 and figure1-—figure supplement 2 are shown for different lengths of time, different carbon sources (electron donors), and different media. To make it easier for the reader to compare, it would help to have the name of the medium as a title for each current density plot. To judge whether riboflavin truly increases current, we should compare figure 1c to figure 1-—figure supplement 2D. In figure 1c, with no riboflavin added, current increases to ~120 uA/cm2 at around 1 day. In the supplemental figure, this increase is not seen until after riboflavin is added. Without a control, it is hard to know whether this increase was dependent on the riboflavin addition.

Line 154: Please define homofermentative.

Line 160: "the percentage of" should be removed

figure 2-—figure supplement 2C: I am confused about how this experiment controls for the possibility that reduction of soluble iron (i.e., the ferric ammonium citrate ), which wouldn't necessarily require EET, contributes to the higher levels of ferrous iron observed when DHNA and ferric ammonium citrate are both added. Maybe a control with no ferrihydrite added would tell us whether ferric ammonium citrate is being reduced.

Line 194: more background should be provided for the FLEET-inducing conditions

Lines 304-309: More background information is needed to understand the motivation behind these experiments. Were oxygen and nitrate available as electron acceptors?

Line 305: "…needed for PMF generation in aerobic…"

Line 414: "previous results"; does this refer to results in this paper or a prior publication? If a prior publication, it should be cited here.

---

## [Author Response]

Essential revisions:1) The methods and results of metabolite analyses are not sufficiently described. Further explanation is required if conclusions are to be made about yield, fermentation, and physiology.

We thank the Editor and Reviewers for raising these important concerns. We revised the Materials and methods and Results sections to more comprehensively describe the methods and results of the metabolite analyses. Within the Results, we included new data (Figure 5 and Supplement file 1) showing the levels of formate and pyruvate under extracellular electron transfer and open circuit conditions. We also clarify that we included standards for a number of metabolites, but that our measurements show that these compounds were absent or sufficiently low to not be within the range of detection.

Second, we also now provide a more complete picture which allows us to make quantitative conclusions about yield, fermentation, and physiology during EET in *L. plantarum*. Our new data on formate and pyruvate levels allowed us to track carbon and electrons at ~80% and ~85%, respectively. With this information, we show that EET increases the yield of ATP and fermentation products per mole of mannitol (Y_mannitol,_ Y_fermentation_ in Table 1) by ~1.75x. This result indicates that EET enables more efficient energy conservation and greater mannitol fermentation in *L. plantarum.* The better accounting of electrons reveals that *L. plantarum* uses an ~2:1 ratio of endogenous to exogeneous electron acceptors during EET*.* Interestingly, the yield of biomass per ATP and per mannitol (Y_ATP,_ Y_mannitol_) decreases by ~1.5x, showing that while EET triggers faster, more efficient energy conservation, these catabolic processes are also less coupled to anabolic processes. Thus, the revised manuscript provides an improved, multifaceted view of this hybrid metabolism.

2) Some claims are overstated and need to be revised or, better yet, backed up with more experimental data.

We revised claims to more precisely reflect our findings and, for other claims, provided additional experimental data to further support them as follows:

(1) We provide new data on formate and pyruvate and the implications of those data on yield of ATP and fermentation products (discussed in our response to point 1 above).

(2) We provide new evidence that *L. plantarum* performs EET with physiologically-relevant (low) concentrations of DHNA (concern raised by Reviewers #2 and #3). DHNA is present at a range of concentrations (0.09-0.5 ug/mL) in foods and other environments inhabited by *L. plantarum*. Our new data (Figure 1—figure supplement 3) shows that *L. plantarum* supplied with 0.010 ug/mL DHNA, which is on the lower end of this range, still produces significant current and iron reduction. Additionally, we now show that this DHNA-dependent EET causes a significant decrease in pH (Figure 1—figure supplement 3), indicating EET affects the physiology of *L. plantarum*.

Lastly, while these data indicate that physiologically relevant levels of DHNA have physiological effects on *L. plantarum* in laboratory culture media, we also qualify our conclusions to explain that the extent and physiological impact of EET will depend on the environmental and nutritional conditions of the niche *L. plantarum* is inhabiting (lines 153-162).

(3) More background information is provided on the rationale and need for measuring oxidation-reduction potential (ORP) (concern raised by Reviewers #1 and 2) (lines 213-217). Appropriate statistical tests are also now applied to the ORP data. Lastly, new data are included (Figure 3B and Figure 3—figure supplement 1) showing that DHNA supplementation is required for robust ORP reduction by wild-type *L. plantarum*, further linking EET activity to its ability to modulate extracellular ORP during fermentation.

(4) Reviewer #3 suggested that the claim that EET proceeds independently of respiration may be overstated. We now explain our current understanding and the limits on that understanding. Our results show EET proceeds via a respiratory protein, but ATP generation during EET is mainly produced by substrate-level phosphorylation, not oxidative phosphorylation. We also acknowledge that EET could involve other as-yet unidentified respiratory components (lines 478-485 and 504-5) and that we cannot exclude ATP generation by oxidative phosphorylation (lines 482-485).

Reviewer #1 (Recommendations for the authors):1) The only weakness: The methods and results of metabolite analyses are not sufficiently described. This requires explanation if conclusions are to be made about yield, fermentation, and physiology.The paper provides strong evidence that electron diversion to external acceptors is an electron sink. However, while quantitative conclusions are made about fermentation, only ~30% of the mannitol can be accounted for in the EET condition that is the focus of this paper. Put another way, the paper is about where electrons go, but the majority of the electrons are missing.Clarify if this strain can excrete products such as sugar polymers, sugars, pyruvate, formate, glycerol, acetoin, or butanediol. State if standards for these were included in the product analysis (an RI detector can pick up nearly all of these). Explain why formate isn't present (was it not measured?), as this would allow determination of flux through PFL vs. a pyruvate excretion, a carbon sink of acetoin, or butanediol. Could missing carbon be due to excretion of a partially oxidized sugar like fructose (which would also show up on an RI detector), or accumulation of storage polysaccharide?

We thank the Reviewer for the Reviewer’s strong endorsement of our manuscript and clear recommendations for improvements.

In the revised manuscript, we provide additional data showing where ~80% of the carbon goes under EET conditions (Supplement File 1) and where the majority of the carbon goes under open circuit conditions. With this more complete information, our balances account for between 77-96% of the electrons (see Supplement file 1). These new data more completely support our conclusion that EET increases fermentation product yield, metabolic flux, and environmental acidification.

A) Address these questions in the section around line 299; where fermentation products are mentioned as being present or absent. What was looked for, but not found (lactate yes, succinate no), vs which were not looked for, (pyruvate? Formate? Acetoin, etc). This makes it clear that carbon is not missing, just that the methods were unable to detect it.

We agree with the Reviewer on this recommendation to provide more clarity on the methods and results of the metabolite analyses. As a result, we have included additional information in the Materials and Methods (lines 712-720) and in the Results (lines 323-325 and 328-335). As described in detail below, we can account for ~80% of the carbon and electrons under EET conditions, which captures a majority of both the carbon and electron flow.

With our HPLC methods, we were able to detect the following compounds using standards: acetate, formate, pyruvate, malate, lactate, succinate, oxalacetate, fumarate, ethanol, acetoin, butanediol, mannitol, glucose and fructose (now indicated on lines 712-720 in Materials and Methods). Despite being extensively studied, *L. plantarum* have not been reported to make storage polysaccharides, and thus it is very unlikely that storage polysaccharides could account for the remaining missing carbon.

We detected large and significant changes in the concentration of main fermentation end-products, acetate, lactate and ethanol (Figure 5). We detected significant, but small changes, in pyruvate and formate production (now shown in Figure 5—figure supplement 2). We did not detect fructose, malate, glucose, oxaloacetate, fumarate, acetoin or 2,3-butanediol in the samples. We have included this new information in the revised Results section (lines 333-334). We also now include carbon in biomass as part of the carbon recovery in an updated Table 1 and updated Table S1. (We also revised other portions of Table S1 to correct a prior error.)

Taken together, these updated measurements increase the carbon recovery observed under both EET and OC conditions. Under EET conditions, ~80% of the carbon is recovered in fermentation products, while under OC conditions, we can account for ~55% of the carbon. We did not measure CO_2_ production because our bioelectrochemical reactors are not gas tight. Thus, we suggest the missing carbon is likely due to CO_2_ production, but we cannot rule out production of other unidentified metabolites.

B) Table 1 and all other tables making calculations need significant qualifiers-- since NADH flux is guessed from mannitol disappearance (the assumption of 3 NADH, which would not be true if sugars, glycerol, or pyruvate were made), and NADH fates + ATP yields are based on a small subset of fermentation products detected. Statements like line 347 "EET allowed cells to produce more fermentation products per mol mannitol" can't be made. Nor can one conclusively say what % of electrons went to electrodes, nor what the YATP is. If anything, the fact that the mix of lactate, acetate and ethanol do not shift dramatically to acetate production (which is typically the result of any system with an NADH sink, such as interspecies hydrogen transfer), is surprising and indicates something unexplained is going on.

We agree with the Reviewer that the qualitative observations are quite surprising and that our quantitative calculations need to carefully state what uncertainties remain. We also believe that our calculations need to be considered in the context of the prior, extensive literature on *L. plantarum* fermentation (McFeeters and Chen, 1996 – *Food Microb*; Teusink *et al.,* 2006 – *J Biol Chem):,* (Teusink et al., 2009 – *Plos Comp Biol*).

Additional context to the changes made in the manuscript: our improved data (Table 1) now aligns with existing literature on *L. plantarum,* allowing us to make quantitative assessments of changes in ATP and NADH. Specifically, our findings now have carbon recoveries (~80%) within 10% of previous studies of *L. plantarum* metabolism (~90-105%, see references below). Additionally, our studies probe the same spectrum of metabolites found in these prior studies. with the sole exception of CO_2_ (CO_2_ cannot be measured using our existing bioelectrochemical reactors, since they are not gas-tight). Because CO_2_ production does not regenerate NADH or produce ATP, the lack of CO_2_ accounting will not affect ATP or NADH balances. Thus, the conclusions we draw about quantifying ATP generated by substrate-level phosphorylation aligns with other quantitative conclusions from prior work on *L. plantarum* metabolism (Dirar and Collins, 1972 – *J Gen Microbiol*).

In response to the Reviewer’s comment, we now use two different sets of assumptions to NADH generation and present the resulting values as a range of NADH regeneration. As described in Supplementary file 1, Method 1 uses the concentration of metabolites to calculate NADH production, essentially assuming that all NADH production stems from production of metabolites. Because it does not include NADH produced from mannitol consumption directed towards biosynthesis or other unidentified fermentation products, Method 1 likely underestimates total NADH produced, leading to higher values of NADH regeneration. Conversely, Method 2 estimates the NADH produced from mannitol consumption, essentially assuming that NADH is produced whenever mannitol is consumed. Since mannitol consumption may lead to biosynthesis or unidentified fermentation products without NADH generation, Method 2 likely overestimates the total NADH produced, leading to lower NADH regeneration values. In light of this uncertainty, we now use both Methods and present a range for NADH regeneration.

Again, the lack of a carbon balance takes nothing away from qualitative conclusion that mannitol electrons are diverted to EET, or the strength of the EET evidence. The electrons at the electrode or in the iron are real and countable, this just limits the quantitative statements that can be made.

We recognize that our measurements do not account for all the carbon, however, in the revised manuscript, these quantitative uncertainties are now limited to ~20% because of the much improved carbon and electron balance. To address the Reviewer’s concern, we have now indicated the limited extent of these uncertainties in the manuscript (lines 363-365).

2) L136; Reported Geobacter current densities easily reach 1000 µA/cm2 for flat surfaces, not 100-200.

Following Reviewer #2’s suggestion, we now compare the current production of different species normalized per g of protein biomass (line 140-145). This normalized value is a better metric to compare EET at a cellular level among different species.

3) Figure 3B: Many questions about this experiment. It is the only time it appears, yet is used as a quantitative way to compare mutants.a) State if the iron(III) in the medium was reduced by mutants during this assay-for redox potential to drop equally suggests that it was (which contradicts 3A).b) Provide the final pH during these experiments-- pH can affect E{degree sign}'.c) Lack of DHNA is always used as a reliable negative control in the other experiments, provide any data that shows if the addition of DHNA is essential for the mV drop in this assay, so we know it is also an assay of EET.Lacking better evidence/controls, this is one of the weaker bits of data in the main body, especially next to the more robust Fe/current data. Appropriate for supplements, but not conclusive as is.

We now provide background and justification for our use of ORP experiments (lines 213-217). Extracellular redox potential is also referred to as oxidation-reduction potential (ORP), which is the ratio of all oxidative to reductive components in the environment (Killeen *et al.*, 2018 – Am. J. Enol. Vitic.). Redox reactions are prevalent in food fermentations (see Hansen, 2018 – Curr. Op. Food. Sci. for a comprehensive list) and ORP directly influences the outcome of fermentations with lactic acid bacteria (Brasca *et al.*, 2007 – J. App. Micro., Olsen and Pérez-Díaz 2009 – J. Food Sci.).

A) Already after one hour incubation in mMRS there was a significant difference in ORP between the wild-type and *ndh2* deletion mutant. This change was also observable for the *pplA* deletion mutant after three hours incubation. ORP differences between the wild-type and the two mutant strains persisted for the duration of the experiment. Additionally, we note that other factors besides EET will affect ORP. Although our new data show that DHNA, and therefore EET capacity, is a major driver of ORP reduction, *L. plantarum* and related bacteria (e.g. *L. lactis*) possess NADH oxidase and therefore are able to reduce O_2_, thereby also contributing to ORP reduction (Tachon *et al.*, 2010 – App. and Env. Micro). Hence, although we expect that iron was reduced by the wild-type *L. plantarum* strain, it was not the only redox active compound. We revised the text (lines 219-228) and Figure 3B to clarify these distinctions.

B) pH values along with the experimental data collected in response to comment C (below), have been added to a new Figure 3 – supplement figure 1. The culture medium pH was identical between the wild-type and mutant strains. Therefore, it is very unlikely that pH alone accounts for the differences in the observed ORP values.

C) We appreciate the Reviewer’s suggestion of including a control group lacking DHNA. We performed that experiment and the results are included as Figure 3 – supplement figure 1. As expected, the absence of DHNA during *L. plantarum* growth led to significantly higher ORP values.

4) An idea regarding the ∆pplA phenotype-the two conditions require EET, but iron oxides should be lower potential acceptors than the anode which is poised at a high driving force of >+400 mV. The anode is plenty high to oxidize something like DHNA (E{degree sign}' ~70 mV) or chelated metals. PplA could be more necessary near the low potential of flavins. The supplied data can't be used to compare the effect of specific riboflavin concentrations on iron reduction rates (only extents) vs current production rates, but it might be available.

We agree with the Reviewer that *L. plantarum* likely uses different mechanisms and proteins for EET depending on the redox potential of the extracellular electron acceptor. Indeed, PplA uses FMN as its redox co-factor (E_0_*’* ~-150 mV and -350 mV), and so it may only reduce electron acceptors with lower potentials (for ferrihydrite E_0_*’* ~ -100 mV -+100 mV). We also believe that the mechanisms and proteins for EET depend on the prior growth conditions for *L. plantarum* before addition of the extracellular electron acceptor. In the revised manuscript, we provide these hypotheses and indicate that testing them is the subject of future work in our laboratories (lines 551-553).

5) Line 311, More data would be needed to use succinate production as evidence for or against respiration. A mol of pyruvate could be excreted as succinate due to TCA cycle oxidation (indicating respiration), manifesting as excess electrons relative to the amount of succinate produced (as additional pyruvate would have to be used to make OAA). Conversely, succinate could be due to electron disposal via a reductive pathway (indicating fermentation). Lacking good carbon or electron recovery, neither can be ruled out, though the non-respiratory reductive pathway is most logical.

While the Reviewer’s reasoning is sound, *L. plantarum,* like other lactic acid bacteria, only possesses the reductive branch of the TCA cycle; it does not possess an oxidative branch (Tsuji et al., 2013 – *Enzyme Microb Technol*). Thus, we can rule out the possibility of TCA cycle oxidation, leaving the reductive branch as the only way to produce succinate. In agreement with the prior literature, we detected only trace amounts of succinate amongst the many TCA cycle metabolites we looked for: succinate, fumarate, oxaloacetate, citrate, malate. Thus, we are confident that the metabolic flux through the reductive branch of the TCA cycle was marginal, if present. In the revised manuscript, we now explain that *L. plantarum* does not possess an oxidative branch of the TCA cycle and that our measurements were sensitive to, but did not detect, these other TCA cycle intermediates (lines 308-310 and 312-314).

6) Table 1/line 403-: The low YATP in cells using EET is a possible 'different energy conversion strategy'. (Side note- an energy spilling/Russell mechanism would be an 'ATP-dissipating', not -accumulating). I'd suggest uncoupling of acid production from growth by low pH seems likely. In most strains, simple glycolytic pathway functions at low external pH, but complex protein synthesis and replication processes become inhibited, ATP from substrate turnover is needed to maintain pmf and internal pH, and high internal organic acids become toxic. Citable yield/pH data can be hard to locate as it comes from the 80's in the age of calling every lactic acid bacterium a Streptococcus, but since the non-EET cultures at neutral pH got close to the Bauchop and Elsden YATP, and EET cultures at pH 4 were a fraction of this, these data look like common behaviour of many lactic acid bacteria. In future comparisons, pH control could help estimate energy yields from EET. The first paper I found in a quick search to illustrate pH effects was: Russell and Dombrowski, DOI: 10.1128/aem.39.3.604-610.1980.

We thank the Reviewer for these insights. We removed the energy-spilling reference from the manuscript. Since our data shows that final biomass is not affected by EET whereas energy conservation is, it seems likely that growth is not strongly coupled to energy conservation under EET conditions. We strongly agree with the Reviewer’s suggestion that the low pH under EET conditions may uncouple ATP generation and growth. We also hypothesize that this ATP produced from substrate-level phosphorylation might be used to create proton motive force to maintain intracellular pH and biosynthetic functions. We have included this hypothesis in the main manuscript (lines 406-409).

7) The value for Geobacter ATP/substrate yield in Table S2 seemed very high. Mahadevan says Geobacter produces 0.5 mol ATP/mol acetate when using extracellular acceptors (3x higher when using internal acceptors like fumarate). If I understand the adjustment for 6-carbon compounds, would this make the value for Geobacter using EET to be 1.5 mol ATP/mol 6-C substrate?

We thank the Reviewer for this correction. We incorrectly provided the thermodynamic boundary for the ATP/substrate yield from the referenced study. We now provide the value of 1.5 mol ATP/mol 6 carbon substrate (0.5 mol ATP/mol acetate) (Mahadevan et al., 2006 – *Appl Environ Microbiol*) in Supplement File 2.

Reviewer #2 (Recommendations for the authors):Line 21 – acronyms like this make our field more challenging to understand and are unnecessary. Does the work presented actually support 'energy conservation'? Growth seems to occur after EET activity takes place rather than with it as discussed below.

We agree with the Reviewer that a broad scientific audience will find the FLEET acronym confusing and have eliminated it from the manuscript.

The Reviewer also questions whether our findings indicate that EET supports energy conservation in *L. plantarum*. In biochemistry, energy conservation is defined as the conversion of light or chemical energy into cellular energy in the form of ATP by an organism (Russell and Cook 1995 – *Microbiol Rev*). In rapidly dividing cells, energy conservation, a catabolic process, is associated with growth, an anabolic process. However, catabolism need not be coupled with anabolism, and energy conservation can still occur in cells that are not rapidly dividing, e.g. in resting cell suspensions (Russell and Cook 1995 – *Microbiol Rev*).

In *L. plantarum*, EET increases the rate at which chemical energy (in mannitol) is converted to cellular energy (in ATP) per cell by ~1.75-fold (Figure 5C), indicating that EET increases the rate of energy conversation per cell. EET increases the amount of cellular energy obtained per chemical energy (Y_mannitol_) (Table 1) by ~1.75-fold, indicating that EET is also a more efficient energy conservation strategy in *L. plantarum*.

To understand how the anabolic process of energy conservation is coupled to the catabolic process of growth, we can compare changes in ATP to changes in biomass. At the start of EET, increased ATP levels are accompanied by a significant shortening of lag phase (Figures 1C, 4C). Since lag phase is the period microorganisms use to accumulate ATP for exponential growth, this shortened period suggests catabolic and anabolic processes are well coupled at this stage. However, by stationary phase, the biomass yield (Table 1) decreased under EET conditions, suggesting a much poorer coupling of catabolic and anabolic processes.

Line 27 – 'fermentation yield' referring to biomass yield or fermentation product yield?

The text was revised to clarity that ‘fermentation product yield’ is being discussed here.

Line 29, Line 45, Line 75, Line 86, Line 479 – blending fermentation and respiration has been known for over a decade in Shewanella (see Hunt et al., 2010, JBact). The description on Line 517 regarding the major finding from this reference is incorrect. PMF is not used by S. oneidensis anaerobically to generate ATP as demonstrated by the robust anaerobic growth of an ATP synthase mutant.

First, our work was very much inspired by the observations in *S. oneidensis* and other work, so we apologize for the inadvertent omission of the Hunt *et al.* J. Bacteriology 2010 reference in the introduction and mis-stating of the ATP synthesis result. In the revised manuscript, we removed the reference to *S. oneidensis* as a well understood respiratory microorganism (line 45), have replaced reference 14 with the Hunt *et al.* J. Bacteriology 2010 (our original intention) (line 80), removed the word recent when describing these findings (line 85), and revised the discussion to correctly describe that fumarate reduction supports the creation of ATP via substrate level phosphorylation, not PMF in *S. oneidensis* (line 523).

Second, we do not claim that our work is the first observation of a hybrid metabolism that blends fermentation and respiration. Rather, this work quantitatively elucidates a different blending of fermentation and respiration that both contrasts with and expands upon previous work. As a counterpoint to the prior observation that substrate-level phosphorylation is the major energy conservation strategy during anaerobic fumarate reduction in a non-fermentative organism (*S. oneidensis*), our work demonstrates that a blending of fermentation and respiration occurs during reduction of solid electron acceptors in a primarily fermentative organism (*L. plantarum*) (lines 28-30, 525-527). Going beyond prior qualitative observations, we show that this hybrid metabolism leads to ~1.75x-more efficient and ~1.75x-faster energy conservation (Y_mannitol_, mannitol flux), but an overall ~1.5-fold weaker coupling between anabolism and catabolism (lower biomass yield). Additionally, we discover that this hybrid metabolism substantially affects the intracellular redox state and show redox change arises from a ~2:1 use of endogenous to extracellular electron acceptors. This fuller discussion is included in the manuscript (lines 525-535).

Line 61 – implied here is that LAB are auxotrophic for heme and menaquinone, but should be explicitly stated.

The text was revised to incorporate the Reviewer’s suggestion (lines 66-68).

Page 8 (top) – the comparison to other organisms is not particularly helpful unless current per unit cell or unit biomass is quantified, nor does it seem particularly relevant to the overall theme of this manuscript.

We agree with the Reviewer that current should be normalized per unit biomass, and we believe that this normalized value provides context for how significant EET is as a metabolic activity. In the revised manuscript, we normalize the current from *L. plantarum* per unit protein and compare this value to reported values for *Geobacter sulfurreducens* and *Shewanella oneidensis* MR-1 (lines 140-147).

Line 152 – predicted to encode

The text was revised to state the predicted protein encoded by *pplA*.

Line 182 – predicted to encode

The text was revised to state the predicted protein encoded by *ndh2*.

Line 183 – under the condition tested.

The text was revised to incorporate the Reviewer’s suggestion.

Line 200 – isn't this already known from the Listeria work?

Prior work in *ListeriaListeria monocytogenes* (Light et al., 2018) showed that iron reductase activity was associated with the presence of PplA and Ndh2. However, as a respiratory species, *L. monocytogenes* has a metabolic strategy that is distinct from lactic acid bacteria. Thus the finding that iron reduction is associated with PplA and Ndh2 in lactic acid bacteria is new. Additionally, our work shows for the first time in any bacterium that *ndh2* and *pplA* are upregulated during growth in the presence of DHNA and iron.

Line 206 – unclear what 'extracellular redox potential' is, unclear to me if any differences are significant in Figure 3B. I guess this is a measurement of Fe(II) / Fe(III) in the medium?

Extracellular redox potential is also referred to as oxidation-reduction potential (ORP), which is the ratio of all oxidative to reductive components in the environment (Killeen *et al.*, 2018 – *Am. J. Enol. Vitic*.). The text has been revised to provide a definition of ORP and this additional background to highlight its importance (lines 213-217).

Significant differences in ORP were found between the wild-type and mutant strains. Figure 3B was revised to now at which time points there ORP values differed between the wild-type and mutant *L. plantarum* strains.

The Reviewer is correct in that iron (II) / iron (III) complexes influence ORP, but there are many other drivers of ORP reduction in the MRS medium, such as LAB consumption of oxygen with NADH oxidase (Zotta *et al.*, 2017 – *J. App. Micro*) (see also response to Reviewer #1). This important facet of the experiment is now described in the manuscript (lines 219-223).

Figure 3D – did the authors include a strain that lacked the cluster?

No, we did not probe a strain that lacked the FLEET locus because this experiment was designed to probe the importance of specifically PplA for current production. Rather, we investigated strains lacking the FLEET locus using our Fe^3+^ reduction assay since this experiment allows testing a larger variety of species at the same time. We believe that the Reviewer’s suggestion would be a useful expansion of our experiment and will include this in our future studies.

Line 252 – unclear to me how ICP-MS can distinguish between iron inside a cell and associated with the cell surface.

We appreciate the Reviewer’s question here. While ICP-MS cannot spatially resolve iron that is intracellular vs. extracellular, we removed surface-associated metals by washing the cells twice in PBS before performing ICP-MS. This washing procedure has been shown in other bacteria to remove loosely bound metals (see Kumar *et al.*, 2020 – Nat. Sci. Reports and Arauz *et al.*, 2008 – J. Haz. Materials). An ICP-MS section was added to the Materials and methods (lines 811-821) to clarify these points.

Line 256 – have other metals been tested? Frequently manganese substitutes for iron in enzymes from non-iron dependent bacteria. Manganese oxide is also a known substrate for EET in Geobacter and Shewanella and could be nutritionally significant to LAB.

For simplicity, we focused on intracellular iron concentrations within the paper, but indeed ICP-MS was also able to detect other metals in *L. plantarum* cells. We now include the other findings from the ICP-MS analyses in Figure 4—figure supplements 1 and 2. Although quantities of barium and calcium were significantly lower and higher in the *ndh2* mutant compared to the wild-type strain, respectively, these differences were not observed for wild-type *L. plantarum* when grown in EET-simulating culture medium (mMRS with DHNA and iron supplementation) compared to the mMRS control medium. In addition, these two metals are not expected to be redox-active metals for EET.

Line 262 – does this mean a large amount of the cells that grew are now dead?

The Reviewer makes an interesting observation. Indeed, the combination of a 4-fold increase in dry weight and only 2-fold increase in viable cells may indicate that a fraction of the biomass was dead or not metabolically active enough to form a colony at the time point sampled. In support of this idea, our measurements of cell density by OD_600_ suggest that cells reach stationary phase shortly after reaching maximum current density. Additionally, shortly after maximum current density (after day 2, Figure 3C), the rates of mannitol consumption, lactate production, and ethanol production decrease (Figure 4). Thus, we hypothesize that at the point of maximum current density, when dry weight and CFU measurements were taken, the cells are entering stationary phase and a fraction of them are metabolically inactive or dead. In the revised manuscript, we point out that this discrepancy could be due to the increased acid stress on *L. plantarum* resulting from EET condition (lines 337-340).

Figure 4 (and Figure 1) – I don't understand why maximum current output is completely decoupled from when biomass spikes. At 24h (maximum current), basically no growth has occurred. At 2 days and 5 days, when biomass is the greatest (in the WT culture), very little current is produced. This seems like direct evidence that EET is not coupled with metabolism in these bacteria. The ATP measurements were made at 24h, where OD of the mutant is decreasing (thus less ATP would be expected). Same issue with NAD/NADH measurement.

We apologize for introducing confusion here, which stemmed from a lack of clarity about which current density and biochemical measurements were from different experiments. We have clarified in the Figure Legends that the same set of experiments yielded the current density plots in Figure 3 and the metabolite and pH measurements shown in Figure 5. We have also clarified that another set of experiments were used to generate the current density shown in Figure 1C and the ATP and NAD/NADH ratios shown in Figure 4.

Comparison of the data shown in Figure 1C and 4C show maximum current is closely followed by maximum OD. After one day, the current density has reached its maximum, but no significant increase in planktonic biomass was observed. Between day one and day two, lower but significant current is produced, the cell density reaches ~90% of its maximum, and the consumption of mannitol and production of lactate also increases dramatically. Overall, these observations show that current production is followed by increases in growth and metabolic flux. We hypothesize that the delay observed between current production and the boost in growth and metabolites could be the time cells need to accumulate ATP and compounds needed to enter exponential growth (now included on lines 274-278). In future experiments, we seek to more precisely relate current produced and growth by continuously monitoring of *L. plantarum* cell density under EET conditions.

Figure 4C – Lines are undefined in the figure legend. Also, growth should be plotted on a semi-log scale, see: https://schaechter.asmblog.org/schaechter/2018/07/why-you-must-plot-your-growth-data-on-semi-log-graph-paper.html

We have addressed the reviewer recommendation and edited Figure 4C accordingly.

Figure 5 – based on the growth curve shown in Figure 4, the OC culture lags behind and does not reach maximum OD until day 5. What do these measurements look like at day 5 (or 6)? It seems likely that the authors are drawing conclusions from cells in two different states (stationary phase vs exponential phase) rather than focusing on the finding that having an external electron acceptor appears to accelerate the metabolism of cells.

The Reviewer is correct that at the time of ATP and NAD/NADH measurements, the cells performing EET and cells under OC were at different growth stages. In Figure 4, we probed how EET in *L. plantarum* affected growth, ATP and intracellular redox state. To show this connection, we chose to measure ATP and NAD^+^/NADH when EET rate is maximum (maximum current density). These measurements show that energy metabolism is greatly accelerated and changed by EET: cells exhibit higher viability, biomass and ATP accumulation and maintain a higher NAD+/NADH ratio. We agree that performing those measurements at other time points would provide a more dynamic picture of the relationship between EET, energy conservation and redox balancing. Our laboratories will address these questions in future studies.

Line 465 – Description of the experiment setup in 2B does not include addition of flavins for the iron reduction assay, so it is unclear why the authors claim that both flavins and quinones are necessary for EET in diverse LAB species.

We included both flavins and quinones in this statement because we show that flavins are required for EET in Figure 1—figure supplement 1D and Figure 1—figure supplement 2D. Flavins are required as cofactors for Ndh2 (as FAD – see Heikal et al., 2014 – Mol Micro) and PplA (as FMN – see Light et al., 2018 – Nature). *L. plantarum* and several other LAB which we found to be EET-active are auxotrophic for riboflavin (Burgess et al., 2004 – *Appl Environ Microbio*) so riboflavin was already supplemented in the growth medium as the vitamin itself (in CDM) or within yeast extract (in MRS). In the iron reduction assay, cells are metabolically active, but not growing, and would have sufficient holo-Ndh2 and holo-PplA carried over from their growth medium to perform EET. This is supported by new data in Figure 1figure supplement 1E where additional riboflavin supplementation into mMRS did not increase subsequent iron reduction. As such, we decided to not include riboflavin in the iron reduction assay used for Figure 2B.

Line 473 – Iron and electrode reduction is dependent on addition of DHNA. Are these cells likely to ever encounter 20 ug/mL DHNA in the environment? Is the EET activity observed relevant to normal physiology for these organisms, or a lab artifact caused by providing a non-native electron carrier? The argument beginning on line 491 underscores the artificial nature to me.

The Reviewer is correct in stating that it is unlikely for *L. plantarum* to encounter 20 µg/mL DHNA in its native environment. DHNA is found in concentrations of 0.089-0.44 μg/mL in commercial fermented beverages, and under laboratory conditions, microbes can synthesize and secrete DHNA leading to concentrations of 0.37-48 μg/mL (lines 153-156). To test if EET in *L. plantarum* is relevant under these physiological concentrations, we probed whether *L. plantarum* can perform EET with a sub-physiological DHNA concentration of 0.01 μg/mL. Indeed, *L. plantarum* produced significant current density and a significant decrease in pH (new Figure 1—figure supplement 3). Far from being artificial, these results show that the concentrations of DHNA in environments known to support *L. plantarum* growth have significant effects on its physiology and mechanisms to outcompete neighboring organisms.

Line 559 – none of the references here provide any quantitation quinone concentration in the digestive tract. (Relates to the comment above from line 473).

As mentioned above, we have included in the manuscript the physiological levels of DHNA found in environments in which *L. plantarum* is known to reside. To our knowledge, no paper has quantified DHNA concentrations in the gastrointestinal tract. Other forms of quinones have been quantified in the GI tract. For instance, menaquinone concentrations of 5.5±2.7-14.54±0.29 μg/g were quantified in distal colonic contents (Conley and Stein, 1992 – Am J of Gastroenterol), and of 25.3-34.4 nmol/g were quantified from fecal samples (Karl et al., 2017 – Am J Clin Nut). Although we mainly focused on DHNA in this manuscript, it is reasonable to anticipate that *L. plantarum* can perform EET in the GI tract by responding to different forms of quinones due to their shared molecular structures. We will consider testing this hypothesis in our future studies.

Line 637 – is ferrozine toxic to these cells?

While not included in the data for this manuscript, we found that ferrozine does not inhibit *L. plantarum* growth. Therefore, there is no evidence that ferrozine is toxic to *L. plantarum*, nor do we expect that compound to impact the ferrihydrite reduction assay, which takes place over three hours and requires metabolically active, but not actively growing cells.

Line 646 – Were these mutants complemented?

Efforts for mutant complementation were made, but restoration of EET activity was variable, potentially because critical epigenetic regulatory conditions were not provided or because of the challenges of over-expression of membrane-bound proteins (Kang and Tullman-Ercek, 2018 – Methods). We are currently in the process of mutating all genes in the *L. plantarum* EET pathway and are including complementation analysis in order to identify the effects of gene proximity and synteny on EET activity.

Line 653 – the suicide vector should be identified and referenced here.

The text has been revised to specify the suicide vector along with a citation.

General comments:Unclear why Figure 1C shows twice as much current twice as fast as Figure 3C in what appears to be identical conditions?

We observe somewhat variable current production depending on the exact conditions used for inoculation of bioreactors. Although we maintained identical culturing conditions for *L. plantarum* before introducing the cells into bioelectrochemical reactors, the inoculum did show mild growth differences. This can cause significant differences in the electrochemical output (current density). Thus, we suggest the differences in the current density production by the wild-type strain between Figure 1C and Figure 3C result from possible differences on the initial number and activity of cells. Future work will seek to uncover how initial conditions affect EET in *L. plantarum*.

Reviewer #3 (Recommendations for the authors):Figure 1A – though the tubes are aligned with the plot below, it may be helpful for some readers to have labels on this image to clarify the conditions of each.

Numbered labels are now provided in Figure 1A to connect the image and the graph.

Figure 1C – for the top line, the "+DHNA+mannitol" label should be on one line

Figure 1C has been revised to include this suggestion.

Figure 4A – labeling should be more consistent ("anodic" vs. "EET")

We have revised the Figure accordingly and replaced anodic with EET.

Figure 4A – "log" should be removed from the y-axis label

We thank the reviewer for noticing this. We have corrected this typo.

Figure 4C – points on this plot should be connected with straight lines

We have addressed this suggestion accordingly, and we have also plotted Figure 4C in a semi-logarithmic scale following another reviewer’s suggestion.

Figure 5D – axis label is cut off

We thank the reviewer for noticing this and we have corrected this typo.

Figure 5G – FLEET should not be mentioned in the model at all. Instead, Ndh2 should be labeled

We have edited the Figure following the recommendations of the reviewer.

Figure 6A – Label differences between the two kale juice bottles

We apologize for omitting some of the details of the juice preparation in that plot. We have indicated the differences between the two juices in Figure 6D by adding details on the preparation steps.

Authors should also provide more information for the definitions of YATP and the values in Table 1

We have provided the definition of each yield at the bottom of Table 1.

Line 137: Where the authors compare current density capacity for L. plantarum to that of S. oneidensis—are the quoted values for S. oneidensis using (an) endogenously produced shuttle(s), or are they for S. oneidensis + (an) added shuttle(s)? This should be clarified, because L. plantarum has no current density capacity without added DHNA.

We agree with the reviewer that this point should be clarified. The new data we provide from the literature to compare *L. plantarum* vs *S. oneidensis* (mA/mg-protein) corresponds to a study in which flavins were self-secreted. We have also clarified that *S. oneidensis* and *G. sulfurreducens* do not require addition of riboflavin and quinones, unlike *L. plantarum* (lines 141-147). However, because *L. plantarum* possesses genetic elements involved in EET (i.e. *ndh2*) and performs EET with levels of DHNA similar to those found in its environment (additional data are provided to show this) *L. plantarum* does perform endogenous EET.

Authors should offer a mechanism for the conditional pplA phenotype

We briefly discussed the differences in the conditional need of PplA for EET across lactic acid bacteria and now have included an additional hypothesis for the conditional need of PplA in *L. plantarum* (lines 551-535)*.* We have preliminary data that suggest there is a conditional need for PplA in *L. plantarum* to perform EET depending on the culturing conditions and will address this fully in future work.

Switching between the terms “electron donor” and “carbon source” may be confusing for some readers; one term should be used consistently.

We agree with the Reviewer, and we now appropriately use the term “electron donor” throughout the manuscript.

Figure 3—figure supplement 1F is not discussed in the text.

We apologize for this omission. We have included a brief discussion of this supplementary Figure (lines 355-359). This figure now is Figure 3 — figure supplement 2C.

Line 204: typo in “pplA”

The spelling error has been corrected.

The ndh2 mutant should be included in the kale juice fermentation experiments to test the contribution of Ndh2 in this context.

While this is a helpful suggestion, we did not use the Ndh2 mutant in our kale juice fermentation experiment since this experiment focused on the metabolic impact of EET during fermentation. However, we envision future experiments addressing the EET mechanism under ecological conditions will utilize this mutant.

Since we do not have this data, we have revised the Abstract to avoid concluding that *ndh2* is necessary for EET in kale juice (line 27).

Readers may not be familiar with the term “Crabtree-positive”

We agree with the Reviewer and have removed this jargon from the Discussion.

“SLP” should be defined on page 3

We have replaced the acronym ‘SLP’ with the term ‘substrate-level phosphorylation’.

The introduction does not provide enough detail for a broad audience to understand the metabolic pathways being discussed. It would help if some of the detail about L. plantarum metabolism (such as lines 318-321, 553-555) was moved from the results/discussion to the introduction. If L. plantarum is able to grow via respiration (as suggested in lines 304-308), this should also be mentioned in the introduction. The introduction should also be revised to ensure that terms are used unambiguously and to remove inaccurate generalizations. Some examples are given in the comments below.

We appreciate the Reviewer’s suggestions regarding the introduction. Regarding the question of providing additional background on *L. plantarum* and LAB metabolism, the Introduction text (lines 55-71 and 97-104) has been revised to incorporate definitions of homofermentation and heterofermentation as well as stating that *L. plantarum* is primarily homofermentative but also a respiratory-capable LAB.

It is not totally accurate to say that energy per se is generated or created during metabolism. The authors should revise these statements to say that energy is obtained or conserved or that ATP is generated.

We agree with the Reviewer. We have carefully revised the manuscript to consistently use the term ‘energy conservation.’

This paper discusses detailed metabolic pathways and the terms used to describe these pathways should be used carefully and precisely to avoid confusion for the reader. For instance, the authors should precisely define “substrate-level phosphorylation” and “fermentation” and should not use these terms interchangeably. “Substrate-level phosphorylation” refers to a reaction that generates ATP directly, without the use of the proton motive force and the membrane-associated ATP synthase enzyme. “Fermentation” refers to the pathways that restore the NAD+/NADH ratio and allow growth with substrate-level phosphorylation as the sole mechanism of ATP generation.

Per the Reviewer’s helpful suggestion, the Introduction text has been revised to define substrate-level phosphorylation and fermentation (lines 50-54).

Lines 50-51 and 373-374: Some organic compounds can also be respired (for example fumarate).

The Reviewer is correct here. The introduction (lines 41-49) includes a broad definition of anaerobic respiration, which includes using both organic and inorganic compounds. We acknowledge that the original text (lines 373-374) may lead the reader to think that respiration is only associated with the use of inorganic electron acceptors. To avoid this confusion, we now state that another major difference in fermentation and anaerobic respiration is the use of the endogenous versus external organic electron acceptors (lines 377-378).

Lines 51-52 should say “Fermentative bacteria produce ATP solely via substrate-level phosphorylation” because substrate-level phosphorylation also occurs when bacteria are growing by respiration (i.e., in glycolysis).

We have revised this sentence (lines 50-52). Instead of using the word ‘solely’, we prefer to employ the word ‘mainly’ since ATP can also be generated via respiration in some primarily fermentative bacteria, like lactic acid bacteria, under specific conditions.

Lines 71-73: It is unclear how the property of being “exergonic” or “endergonic” is relevant in the context here.

We agree with the Reviewer that providing these details is not relevant and have removed them.

Lines 85-86: “…EET is associated with non-fermentative respiratory organisms…” Does this refer specifically to endogenous EET (i.e., without an added mediator?). This should be clarified because there is a long history of research on fermentation+EET in the presence of added mediators (see next comment).

The Reviewer makes a good point, and we revised the sentence to clarify that we are describing that endogenous EET is mainly associated with respiratory species.

Related to the above point: In the introduction or discussion, the authors should mention or acknowledge other work examining how growth in the presence of an electrode affects fermentative metabolisms. Some examples are:Glasser NR, Kern SE, Newman DK. Phenazine redox cycling enhances anaerobic survival in *Pseudomonas aeruginosa* by facilitating generation of ATP and a proton-motive force. Mol Microbiol. 2014 Apr;92(2):399-412. Doi: 10.1111/mmi.12566.Emde R, Schink B. Enhanced Propionate Formation by Propionibacterium freudenreichii subsp. Freudenreichii in a Three-Electrode Amperometric Culture System. Appl Environ Microbiol. 1990 Sep;56(9):2771-6. Doi: 10.1128/aem.56.9.2771-2776.1990.And other references discussed in reviews such as:Moscoviz R, Toledo-Alarcón J, Trably E, Bernet N. Electro-Fermentation: How To Drive Fermentation Using Electrochemical Systems. Trends Biotechnol. 2016 Nov;34(11):856-865. Doi: 10.1016/j.tibtech.2016.04.009.Vassilev I, Averesch NJH, Ledezma P, Kokko M. Anodic electro-fermentation: Empowering anaerobic production processes via anodic respiration. Biotechnol Adv. 2021 May-Jun;48:107728. Doi: 10.1016/j.biotechadv.2021.107728.

In the original Discussion, we state that fermentation can be modulated using electrodes and cite the Moscovitz reference. However, we acknowledge this statement was brief and have expanded this discussion and included the Vassilev and Emde and Schink references in the revised manuscript (line 594 and lines 495-496). We also included the Glasser et al. reference (lines 93-96) as an example of a primarily respiratory organisms that also ferments performing EET.

Comparing Figure 1A to Figure 3A, the extent of Fe3+ reduction seems quite different. Were the conditions the same for these two experiments? If so, an explanation should be added to the text.

The conditions differed between Figures 1A and 3A. The iron reduction assay results shown in Figure 1A were obtained for *L. plantarum* after growth in mMRS. The results shown in Figure 3A were obtained after *L. plantarum* growth in mMRS supplemented with DHNA and ferric ammonium citrate. The inclusion of DHNA and ferric ammonium citrate in the mMRS growth medium was found to result in increased *L. plantarum* EET activity (data previously in Figure 2—figure supplement 2C). To clarify the rationale for the change in mMRS growth medium, we moved that supporting data to Figure 1—figure supplement 1 and revised the Results section to better describe these results and how they influenced our experimental methods (lines 123-133).

Lines 104-106: Can DHNA itself be used as an electron shuttle? This should be addressed.

Yes, DHNA can be used as an electron shuttle (Mevers, 2018 — *eLife*; Glasser NR et al., 2017 – Annu Rev Microbiol). DHNA can also be synthesized into menaquinone or demethylmenaquinone, which can transfer electrons in an electron transfer chain. However, our current data does not allow us to assess which of these roles DHNA is playing. Future work will elucidate the role of DHNA and flavins in EET in *L. plantarum*.

Line 137: Where the authors compare current density capacity for L. plantarum to that of S. oneidensis—are the quoted values for S. oneidensis using (an) endogenously produced shuttle(s), or are they for S. oneidensis + (an) added shuttle(s)? This should be clarified, because L. plantarum has no current density capacity without added DHNA.

We revised the comparison of EET among species to indicate that, unlike *L. plantarum*, *Shewanella oneidensis* and *Geobacter sulfurreducens* can synthesize riboflavin and quinones and do not require the addition of either for EET activity (lines 146-147).

Line 139: “current production was not dependent on the carbon source or growth medium, and current increased after supplementation of riboflavin in the growth medium”. Both of these claims are hard to evaluate because the plots in figure 1 and figure1—figure supplement 2 are shown for different lengths of time, different carbon sources (electron donors), and different media. To make it easier for the reader to compare, it would help to have the name of the medium as a title for each current density plot.

We acknowledge the lack of clarity here and have revised the text to more accurately convey our point (lines 148-152). Our intention is to qualitatively, rather than quantitatively, evaluate current production with different media. We have followed the Reviewer’s suggestion and added a title for each plot of Figure 1—figure supplement 2 to make it easier for the reader to understand the medium used.

To judge whether riboflavin truly increases current, we should compare figure 1c to figure 1—figure supplement 2D. In figure 1c, with no riboflavin added, current increases to ~120 uA/cm2 at around 1 day. In the supplemental figure, this increase is not seen until after riboflavin is added. Without a control, it is hard to know whether this increase was dependent on the riboflavin addition.

We agree with the Reviewer on this point. We provide additional data in Figure 1—figure supplement 2D that includes a control in which riboflavin was not added. This control shows that the increase in current is due to the addition of riboflavin.

Except for Figure 1—figure supplement 2D, all the experiments performed in bioelectrochemical reactors contain riboflavin in the media (as part of the vitamin stock solution in the CDM). This is now explained in the manuscript text (lines 148-152).

Line 154: Please define homofermentative.

The Introduction text has been revised to include definitions of both homofermentation and heterofermentation (lines 60-64).

Line 160: “the percentage of” should be removed

The legend text of Figure 2 has been revised to incorporate this suggestion.

Figure 2—figure supplement 2C: I am confused about how this experiment controls for the possibility that reduction of soluble iron (i.e., the ferric ammonium citrate ), which wouldn’t necessarily require EET, contributes to the higher levels of ferrous iron observed when DHNA and ferric ammonium citrate are both added. Maybe a control with no ferrihydrite added would tell us whether ferric ammonium citrate is being reduced.

Prior to performing the ferrihydrite reduction assay, the cells were washed twice (in PBS), thereby removing any ferric ammonium citrate that was present in the mMRS growth medium (line 744). Moreover, there was very little color change in the ferrihydrite reduction assay when wild-type *L. plantarum* was grown in mMRS containing ferric ammonium citrate but not DHNA (noting that DHNA was still included in the ferrihydrite assay medium) (Figure 2—figure supplement 2C). Therefore, our data show that ferric ammonium citrate is not contributing to the color formation in the ferrihydrite reduction assay. Additionally, we agree that it is notable how *L. plantarum* exposure to DNHA and ferric ammonium citrate during growth results in an improved capacity for that organism to reduce extracellular ferrihydrite and that the expression of *ndh2* and *pplA* is induced under those conditions (lines 199-206). We are pursuing these questions in studies investigating the transcriptional control and downstream regulation of EET in *L. plantarum*.

Line 194: more background should be provided for the FLEET-inducing conditions

The Results section was revised to clarify and provide background on what was intended by the term “FLEET-inducing conditions” (lines 126-133). For clarity, we have also removed that term from the text and explained directly which culture conditions were used.

Lines 304-309: More background information is needed to understand the motivation behind these experiments. Were oxygen and nitrate available as electron acceptors?

To clarify further the motivation behind this experiment for the reader we have included a more detailed explanation (lines 295-301). Neither oxygen nor nitrate were used as electron acceptors, as their presence would be expected to compete with the anode as an electron acceptor.

Line 305: “…needed for PMF generation in aerobic…”

We have corrected this typo.

Line 414: “previous results”; does this refer to results in this paper or a prior publication? If a prior publication, it should be cited here.

The text has been revised to incorporate this suggestion.